# NK cell-induced damage to *P.falciparum*-infected erythrocytes requires ligand-specific recognition and releases parasitophorous vacuoles that are phagocytosed by monocytes in the presence of immune IgG

**Padmapriya Sekar[1¤a], Sumati Rajagopalan[1]\*, Estela Shabani[2], Usheer Kanjee[2], Marc A. Schureck[3¤b], Gunjan Arora[1¤c], Mary E. Peterson[1,4], Boubacar Traore[5], Peter D. Crompton[4], Manoj T. Duraisingh[2], Sanjay A. Desai[3], Eric O. Long[1]\***

1 Molecular and Cellular Immunology Section, Laboratory of Immunogenetics, National Institute of Allergy and Infectious Diseases, National Institutes of Health, Rockville, Maryland, United States of America, 2 Department of Immunology and Infectious Diseases, Harvard T.H. Chan School of Public Health, Boston, Massachusetts, United States of America, 3 Laboratory of Malaria and Vector Research, National Institute of Allergy and Infectious Diseases, National Institutes of Health, Rockville, Maryland, United States of America, 4 Malaria Infection Biology and Immunity Section, Laboratory of Immunogenetics, National Institute of Allergy and Infectious Diseases, National Institutes of Health, Rockville, Maryland, United States of America, 5 Malaria Research and Training Center, Mali International Center for Excellence in Research, University of Sciences, Techniques, and Technologies of Bamako, Bamako, Mali

¤a Current address: TeraImmune, Germantown, Maryland, United States of America
¤b Current address: Department of Biology, Loxo Oncology at Lilly, San Diego, California, United States of America
¤c Current address: Laboratory of Host-Pathogen Dynamics, National Heart Lung, and Blood Institute, National Institutes of Health, Bethesda, Maryland, United States of America
* srajagop@niaid.nih.gov (SR); eLong@nih.gov (EOL)

**Data Availability Statement:** All relevant data are within the paper and Supporting Information files.

## Abstract

Natural killer (NK) cells lyse virus-infected cells and transformed cells through polarized delivery of lytic effector molecules into target cells. We have shown that NK cells lyse *Plasmodium falciparum*-infected red blood cells (iRBC) via antibody-dependent cellular cytotoxicity (ADCC). A high frequency of adaptive NK cells, with elevated intrinsic ADCC activity, in people chronically exposed to malaria transmission is associated with reduced parasitemia and resistance to disease. How NK cells bind to iRBC and the outcome of iRBC lysis by NK cells has not been investigated. We applied gene ablation in inducible erythrocyte precursors and antibody-blocking experiments with iRBC to demonstrate a central role of CD58 and ICAM-4 as ligands for adhesion by NK cells via CD2 and integrin αMβ2, respectively. Adhesion was dependent on opsonization of iRBC by IgG. Live imaging and quantitative flow cytometry of NK-mediated ADCC toward iRBC revealed that damage to the iRBC plasma membrane preceded damage to *P. falciparum* within parasitophorous vacuoles (PV). PV were identified and tracked with a *P.falciparum* strain that expresses the PV membrane-associated protein EXP2 tagged with GFP. After NK-mediated ADCC, PV were either found inside iRBC ghosts or released intact and devoid of RBC plasma membrane. Electron microscopy images of ADCC cultures revealed tight NK–iRBC synapses and free vesicles

**Funding:** Research at the National Institute of Allergy and Infectious Diseases, National Institutes of Health (NIH), was supported by the Division of Intramural Research, grant AI001238 (S.A.D. and E.O.L.). Research at the Harvard T.H. Chan School of Public Health was supported by NIH grants 5R01HL139337-04 and 5R01AI140751-05, which included salary (M.T.D.). The funders had no role in study design, data collection and analysis, decision to publish, or preparation of the manuscript.

**Competing interests:** The authors have declared that no competing interests exist.

similar in size to GFP⁺ PV isolated from iRBC lysates by cell sorting. The titer of IgG in plasma of malaria-exposed individuals that bound PV was two orders of magnitude higher than IgG that bound iRBC. This immune IgG stimulated efficient phagocytosis of PV by primary monocytes. The selective NK-mediated damage to iRBC, resulting in release of PV, and subsequent phagocytosis of PV by monocytes may combine for efficient killing and removal of intra-erythrocytic *P.falciparum* parasite. This mechanism may mitigate the inflammation and malaria symptoms during blood-stage *P. falciparum* infection.

## Author summary

The parasite *Plasmodium falciparum* is the main contributor to malaria disease that causes more than 600,000 deaths/year, primarily among young children. To control disease, it is critical to understand how clinical immunity develops in people exposed to malaria. Antibodies acquired during repeated malaria exposure are sufficient to protect against disease. One of the mechanisms for protection is antibody dependent killing of parasite-infected red blood cells (RBC) by natural killer (NK) cells, a type of white blood cell. Here we show how NK cells detect and bind to parasite-infected RBC and damage the RBC plasma membrane. We identified two separate NK cell receptors that promote stable interaction only with infected RBC and identified the two proteins that these receptors recognize on RBC. Interactions between NK cells and infected RBC were captured by live imaging, which showed that selective NK-mediated damage to the infected RBC membrane is followed by the release of vesicles ("parasitophorous vacuoles", or PV) that contain parasites. PV are readily detected by antibodies from malaria-exposed individuals and stimulate monocytes to engulf the PV by phagocytosis. This study provides insights into how NK cell activity contributes to reducing parasite load and promotes malaria resistance in people in malaria-endemic areas.

## Introduction

Malaria remains a devastating disease with severe and fatal cases occurring mostly in young children. *Plasmodium falciparum* (*P.f.*) is the parasite responsible for most deaths from malaria and is transmitted by mosquitoes that feed on human blood. Drugs and vaccines available to treat and protect people exposed to malaria transmission are either not effective or affordable enough to eliminate the disease. The human immune system confers clinical immunity (i.e. asymptomatic infection) to most people in malaria-endemic areas who have been repeatedly exposed for several years to malaria transmission [1]. Understanding how this immunity develops and provides protection by restraining *P.f.* induced inflammation while maintaining control of parasite replication could translate into better treatments and vaccines.

IgG from clinically immune individuals is sufficient to protect against malaria disease [2]. IgG specific for different targets at the blood stage such as *P.f.* merozoites and infected erythrocytes could confer protection through neutralization to block invasion of erythrocytes, activation of the complement system to kill merozoites, engagement of Fcγ receptors to activate phagocytosis by myeloid cells and neutrophils, and FcγRIIIA (CD16) to trigger NK cell-mediated antibody (Ab) dependent cellular cytotoxicity (ADCC) [3–7]. Besides blocking or disabling *P.f.* at the blood stage, another requirement for protection is the control of inflammation. Severe malaria cases occur when inflammation is not controlled, which may

lead to cerebral malaria. Regulation of *P.f.*-induced inflammation likely contributes to protection against malaria symptoms [8–11]. Indeed, epigenetic changes observed in peripheral blood mononuclear cells showed that those occurring in myeloid cells of infected individuals had the strongest association with protection [12].

Conventional targets of NK cytotoxicity are tumor cells, virus-infected cells, and activated T cells, which are recognized through multiple ligand–receptor interactions [13,14]. To make up for the lack of antigen-specific receptors, NK cells have an arsenal of coactivation receptors that detect different ligands and function as synergistic pairs [15]. An exception is Fc receptor CD16, which signals on its own for ADCC through multivalent engagement of the Fc fragment of IgG [16]. Lysis of a target cell by an NK cell is a multi-step process, starting with an initial contact that progresses to strong adhesion and formation of an immunological synapse [17]. Integrin αLβ2 (CD11a/CD18, LFA-1) is the dominant adhesion receptor of NK cells. Adhesion is further enhanced by inside-out signals [18] from NK coactivation receptors [19]. Binding of αLβ2 to its ligand ICAM-1 is sufficient to induce signals that promote adhesion and polarization of the NK cell toward the target cell [20]. The nucleus moves to the back of the cell while the centrosome and associated lytic granules move toward the synapse with the target cell [17]. This signal does not induce degranulation. Degranulation is triggered independently of polarization by pairs of coactivation receptors or by CD16 [21]. When signaling pathways for both adhesion/polarization and degranulation are induced, polarized degranulation delivers lytic effector molecules, such as perforin, granzymes and granulysin into target cells [22]. In nucleated target cells, these molecules induce apoptosis through nuclear condensation and mitochondrial depolarization. As erythrocytes lack both nucleus and mitochondria, it is not clear how NK cells lyse them.

Natural cytotoxicity of NK cells is triggered independently of IgG and occurs through several signaling pathways depending on the combination of coactivation receptors engaged with ligands on target cells [21,23]. For example, NKG2D (*KLRK1*, CD314) binding to stress-induced ligands synergizes with 2B4 (*CD244*, SLAMF4) engaged by CD48 (SLAMF2). CD58 (LFA-3), which is expressed on erythrocytes, is the ligand of human receptor CD2. Signaling by CD2 synergizes with immunoreceptor tyrosine-based activation motif (ITAM)-mediated signals, such as those triggered by CD16 [21,24,25].

We have reported that NK cells lyse IgG-coated iRBC selectively without causing bystander lysis of uninfected RBC and inhibit parasite growth and transmission into fresh RBC in vitro [6]. To directly compare iRBC with uninfected RBC in their interaction with NK cells, we used a polyclonal anti-RBC rabbit IgG. Studies with *ex vivo* NK cells revealed that malaria-exposed individuals have a high proportion of adaptive NK cells and that the enhanced ADCC activity of adaptive NK cells was associated with reduced parasitemia and resistance against malaria [26,27]. A subset of adaptive NK cells characterized by the lack of CD56 (*NCAM1*) and higher expression of inhibitory receptors *LAG3* (CD223) and *LILRB1* (CD85j) correlated with reduced parasitemia and protection against malaria symptoms in the subsequent year [27].

The aim of our study was to determine how NK cells mediate antibody-dependent lysis of iRBC and to further investigate the fate of iRBC and the parasite within after NK cell-mediated lysis. What molecules on infected RBC do NK cells recognize and what kind of response is activated? What is the fate of intra-erythrocytic *P.f.* parasite when an iRBC is attacked by NK cells? Here, we identify two separate receptor–ligand interactions that contribute to stable NK–iRBC adhesion and report that initial damage to the iRBC plasma membrane is followed by the release of parasitophorous vacuoles (PV) from ghost iRBC. Furthermore, in the presence of human IgG from individuals resistant to malaria these PV were cleared by monocytes through phagocytosis.

## Results

### NK cells adhere preferentially to *P.f.*-infected RBC and lyse them selectively via ADCC

As tight adhesion with target cells is a prerequisite for perforin and granzyme-dependent lysis by NK cells, we examined conjugate formation between primary human NK cells and *P.f.*-infected RBC (iRBC). A two-color flow cytometry adhesion assay was performed with NK cells [28] after incubation with uninfected or iRBC for 20 min or 40 min. In the absence of poly-clonal anti-RBC rabbit IgG, which binds equally well to uninfected and iRBC (S1 Fig) and acti-vates NK cells by engagement of FcγRIIIA (CD16), very little adhesion occurred with either uninfected or iRBC (Fig 1A). NK cell adhesion to nucleated target cells is generally indepen-dent of CD16 signals [16] and mediated primarily by integrin αLβ2. In contrast, NK cells adhered to iRBC only in the presence of anti-RBC IgG (Fig 1A), suggesting that signals from CD16 provided the inside-out signal required for strong adhesion. Remarkably, NK cells dis-criminated between uninfected and iRBC even in the presence of anti-RBC IgG (Fig 1A). The dependence on IgG for adhesion with iRBC was observed in multiple experiments (Fig 1B), as was the clear difference in conjugate formation with uninfected and iRBC in the presence of IgG (Fig 1C). The strong adhesion of NK cells to iRBC is consistent with the efficient lysis of iRBC through antibody-dependent cellular cytotoxicity (ADCC) [6].

The major adhesion receptor on NK cells, integrin αLβ2, binds to intercellular adhesion molecules (ICAM)-1, -2, and -3, none of which are expressed on human erythrocytes. ICAM-4, also known as Landsteiner-Wiener (LW) blood group antigen, is a divergent member of the ICAM family expressed selectively in erythroid cells and required for erythroblastic island for-mation [29]. Conflicting studies have reported ICAM-4 binding to integrin αMβ2 (CD11b/CD18, Mac-1)[30] or to integrin α4β1 (CD49d/CD29, VLA-4)[31]. RBC also express CD58 (LFA-3), which is the main ligand for CD2 on human T cells and NK cells. To determine what ligand–receptor interactions may occur between iRBC and NK cells, we used a panel of mono-clonal antibodies (mAb) known to block the binding of receptors to their ligand. mAbs to integrin subunits αL (CD11a), αM (CD11b), β2 (CD18), and α4 (CD49d), and to CD2 were tested in a functional assay of NK-dependent lysis of iRBC. We measured hemoglobin (Hb) that was released from iRBC during incubation with NK cells in the presence of anti-RBC IgG and in the presence or absence of blocking Abs (Fig 1D). CD11a and CD49d Abs had no effect on Hb release, whereas CD11b, CD18, and CD2 antibodies inhibited Hb release (Fig 1D and 1E), consistent with a study showing ICAM-4 binding to αMβ2 [30]. Combinations of Abs to CD2 and CD11b or to CD2 and CD18 caused greater inhibition of Hb release. These results suggested that CD2 and integrin αMβ2, but not integrin αLβ2 or α4 contributed to the lysis of iRBC by NK cells in ADCC assays.

### ICAM-4 and CD58 on erythrocytes are required for recognition by NK cells

To obtain more direct evidence of a role for ICAM-4 and CD58 in the lysis of iRBC by NK cells, we turned to a genetic approach. Gene ablation in inducible erythrocyte precursors, such as EJ cells that are differentiated to enucleated erythrocytes, has been a powerful tool to iden-tify molecules that mediate invasion of erythrocytes by *P.f.* merozoites [32,33]. First, we assessed if differentiated EJ cells could be used as a model to study NK–RBC interactions by performing ADCC assays with NK cells. As robust lysis of EJ cells by NK cells, comparable to that of iRBC, was observed in the presence of anti-RBC IgG, we performed the same antibody blocking experiments as those done with iRBC (Fig 1E). Lysis of EJ cells, as measured by Hb release, was inhibited by the CD2 antibody but no other antibody (Fig 1F). Therefore, lysis of

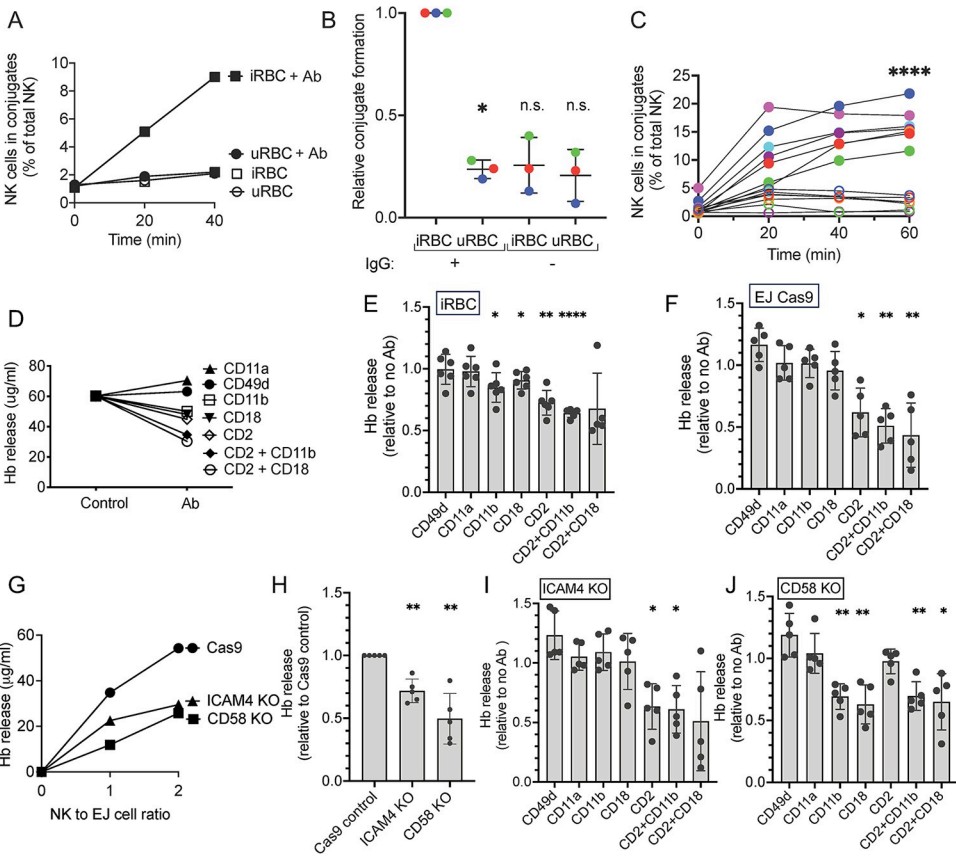

**Fig 1. Ligands on *P.f.*-infected RBC that are required for recognition by NK cells.** (**A**) Conjugate formation between NK cells and uninfected RBC (uninfected RBC) or *P.f.*-infected RBC (iRBC) in the absence (open symbols) or presence (filled symbols) of anti-RBC IgG. NK cells were labeled with Cell Tracker Green and RBC were labeled with eFluor 670. After the indicated times, conjugates were assessed by flow cytometry by gating on double-positive NK-RBC conjugates. (**B**) Relative conjugate formation at 40 min between NK cells and RBC (uninfected RBC and iRBC) at a 1:1 ratio in the presence or absence of anti-RBC IgG. Each color represents a single donor. To compare independent experiments with NK cells from three different donors, conjugates with iRBC in the presence of IgG was set as 1. (**C**) Conjugate formation between NK cells and RBC (uninfected RBC: open circles; iRBC: filled circles) in the presence of anti-RBC IgG. Seven independent experiments with NK cells from different donors are shown. Standard error of the mean: **** p < 0.0001 by one-way ANOVA followed by Tukey's multiple comparisons test. (**D**) Hemoglobin (Hb) released after incubation of *P.f.*-iRBC with NK cells for 1 h at an E:T ratio of 2 in the presence of anti-RBC IgG and blocking antibodies against the indicated NK adhesion receptors. The control is Hb released in the absence of blocking Ab. Hb released was measured by an ELISA assay. A representative experiment is shown. (**E**) Effect of blocking Abs against NK adhesion receptors on Hb released by iRBC during incubation with NK cells as described in (D) in 5 independent experiments with different human blood donors. (**F**) Effect of the same blocking Abs on Hb released from erythroid EJ-Cas9 cells in an assay performed as in (D) and tested in 5 independent experiments with NK cells derived from different human donors. (**G**) Hb released by EJ cells during incubation with NK cells in the presence of anti-RBC IgG. The control were EJ cells transfected with Cas9 alone, which was compared with EJ cells in which the genes *ICAM4* and *CD58* were knocked out independently by CRISPR-Cas9. A representative experiment is shown. (**H**) Five independent experiments with NK cells from different donors, performed as in (G), with EJ-Cas9 cells, EJ-*ICAM4*-KO and EJ-*CD58*-KO cells. To compare separate experiments, Hb released by EJ-Cas9 cells in the presence of IgG was set as 1. (**I**) Effect of blocking antibodies against NK adhesion receptors on Hb released by EJ-*ICAM4*-KO cells in five independent experiments performed as in (H) with NK cells from different donors. Data is shown relative to the Hb released in the absence of blocking Ab. (**J**) Effect of the same blocking antibodies on Hb released from EJ-*CD58*-KO cells in the same assay as in (I) and with NK cells from 5 different human donors: P value (two tailed) * p < 0.05, **** p < 0.0001 by one sample t test with hypothetical value of 1.

differentiated EJ cells, as compared to that of iRBC, was not sensitive to blocking of αMβ2 and more sensitive to blocking of CD2 (Fig 1E and 1F).

Next, we tested EJ cells in which *ICAM4* and *CD58* had been knocked out individually by CRISPR-Cas9 (S1 Fig). ICAM4-KO and CD58-KO cells, along with parental Cas9-transfected cells, were incubated with NK cells in the presence of anti-RBC IgG. Hb release assays revealed a reduced ability of NK cells to lyse these two KO cell lines (Fig 1G). In multiple experiments, lysis of ICAM4-KO cells was approximately 70%, and that of CD58-KO 50% of the lysis observed with parental EJ-Cas9 cells (Fig 1H). This was consistent with recognition of both ligands by NK receptors and with incomplete blocking by Abs to CD11b and CD18. To further examine the contribution of ICAM-4 and CD58 in recognition of EJ cells by NK cells and to validate the specificity of antibodies, we performed antibody blocking experiments. As expected, lysis of ICAM4-KO cells was inhibited by the CD2 antibody but not by CD11b or CD18 antibodies (Fig 1I). Conversely, the CD2 antibody had no effect on the lysis of CD58-KO EJ cells (Fig 1J), providing evidence of the CD2 antibody specificity. Furthermore, the absence of a CD2–CD58 interaction of NK cells with CD58-KO EJ cells unmasked the contribution of ICAM-4, as inhibition did occur with Abs to αM and β2, but not αL or α4 integrin subunits (Fig 1J). This experiment also confirmed the specificity of antibodies to integrin αMβ2, as they did not cause inhibition in the absence of ICAM-4 ligand (Fig 1I). When measured relative to EJ-Cas9 cells, the residual lysis of ICAM4-KO EJ cell in the presence of Ab to CD2 was down to 32%, and residual lysis of CD58-KO EJ cells in the presence of Abs to αMβ2 was down to 42% of the lysis of EJ-Cas9 cells. Residual lysis could be due to incomplete inhibition with Abs or to the existence of other ligands on erythrocytes that contribute to recognition by NK cells. Gene ablation of ICAM-4 and CD58 in EJ cells has established a role for these two ligands in the recognition of erythrocytes by NK cells, and antibody blocking experiments with both *P.f.*-infected RBC and EJ cells were clearly consistent with recognition of CD58 by CD2 and of ICAM-4 by integrin αMβ2.

## Damage to *P. falciparum* during NK cell-mediated ADCC follows the loss of RBC plasma membrane integrity

Efficient and selective lysis of iRBC by human NK cells via ADCC has been reported [6]. As the fate of iRBC and of intra-erythrocytic *P.f.* parasite upon NK-mediated lysis is not known, we developed quantitative assays to examine the integrity of iRBC and parasite. eFluor 670-stained iRBC were gated as glycophorin A+ LFA-1-negative cells (Figs 2A and S2A). Damage to their plasma membrane was detected with phalloidin AF405 (Fig 2B), which enters permeable iRBC and binds F-actin [34]. iRBC were pre-stained with the membrane dye PKH26 (Fig 2B). Parasite integrity was monitored with propidium iodide (PI), which stains DNA in cells that have lost nuclear membrane integrity. After incubation of trophozoite-enriched RBC with primary, freshly isolated human NK cells, phalloidin AF405+ iRBC were detected (Figs 2B and S2). PKH26+ phalloidin-negative are intact iRBC, and peripheral staining with phalloidin identified ghost iRBC, which looked translucent but for the pigment in DIC images (Fig 2B). Only a few iRBC became permeable to phalloidin in the absence of NK cells (Figs 2C, 2D and S2). The fraction of phalloidin permeable iRBC increased with time and IgG concentration (Fig 2C and 2D). A smaller fraction of iRBC became PI+ (Fig 2C and 2E). Virtually all the PI+ iRBC were also phalloidin permeable (Fig 2C), implying that damage to the parasite follows the loss of iRBC plasma membrane integrity. Quantitative analysis of multiple experiments showed that 43.5% ± 5.4% (P <0.0001) of iRBC were phalloidin permeable and 13.5% ± 1.6% (P = 0.0443) were PI+ (Fig 2F), corresponding to 31% of the phalloidin+ fraction.

A granzyme reporter assay was used to test whether NK–iRBC contacts resulted in polarized delivery of granzyme into iRBC targets. To eliminate detection of granzyme activity that

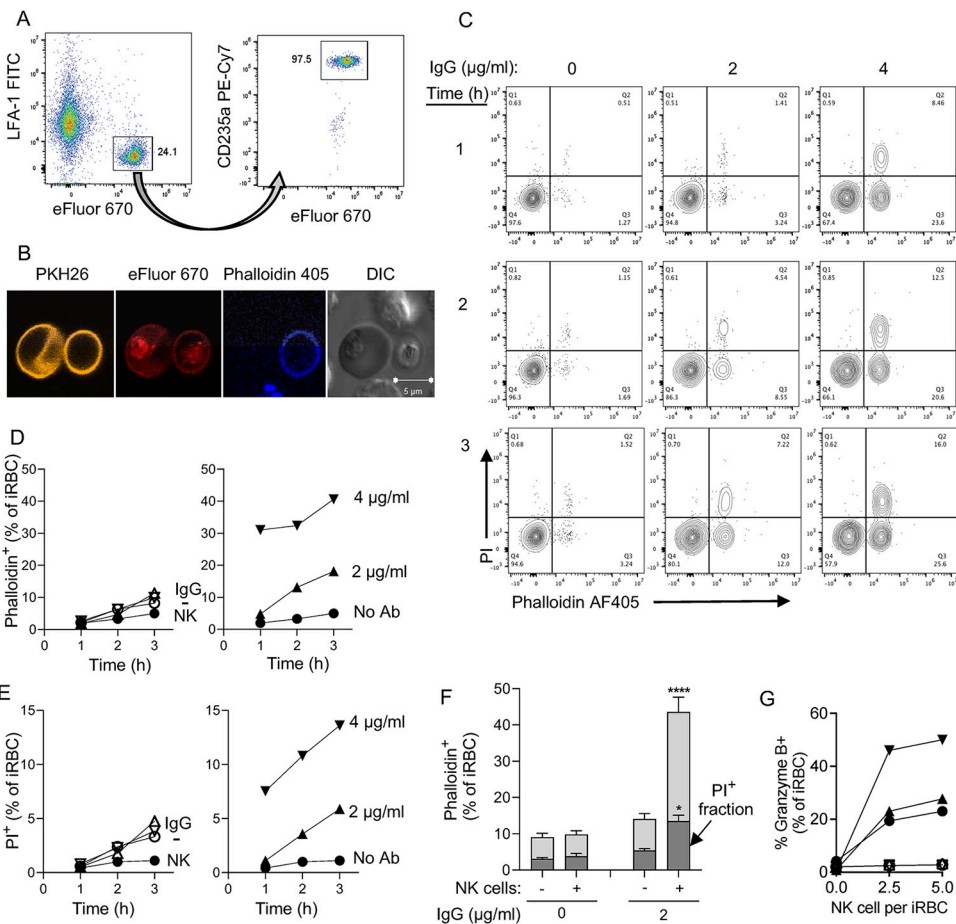

**Fig 2. Sequential damage to *P.f.*-infected RBC and to the parasite during NK-dependent ADCC.** (**A**) Gating strategy to identify primary NK cells and 3D7 *P.f.*-iRBC mixed at a 3 to 1 ratio. iRBC were prelabelled with eFluor 670. After incubation with NK cells, iRBC were stained with an Ab to glycophorin A (CD235a PE-Cy7) and NK cells were stained with an Ab to the integrin LFA-1 (FITC). (**B**) Immunofluorescence image of iRBC stained with the membrane dye PKH26 (yellow) and the primary amine dye eFluor 670 (red) and incubated with NK cells for 3 hours in the presence of F-actin binding Phalloidin-AF405 (blue). The DIC image shows an intact iRBC (left) and a translucent ghost iRBC (right). Two unstained NK cells are visible (top and bottom). (**C**) NK cells and iRBC were incubated at a ratio of 3:1 for up to 3 hours in the presence anti-RBC IgG, as indicated. Damaged iRBC were detected with Phalloidin AF405 and damaged parasites identified with DNA binding of propidium iodide (PI). Results are representative of seven experiments. (**D**) Time-, Ab-, and NK-dependent damage to iRBC detected by phalloidin staining. Control iRBC incubated with nothing (-), Ab alone (IgG), or with NK cells alone (NK) are shown on the left. (**E**) NK-dependent damage to parasites was determined by PI staining. Controls in the left panel are as described in (D). (**F**) Data from seven independent experiments with NK cells from different donors performed as in D and E shown as the proportion of iRBC that became Phalloidin-AF405+ and of intra-erythrocytic parasite damage detected as phalloidin and PI double-positive events (darker shading). Samples were incubated in the presence (+) or absence (-) of NK cells and of 2 μg/ml anti-RBC IgG for 3 hours. Standard error of the mean: * $p < 0.05$, **** $p < 0.0001$ by one-way ANOVA followed by Tukey's multiple comparisons test. (**G**) Granzyme B reporter assay performed with iRBC after incubation with NK cells in the presence of 1 μg/ml IgG. Prior to analysis, NK–iRBC conjugates were disrupted by addition of 5mM EDTA. Residual conjugates were gated out using forward scatter parameters (FSC >600) on the flow cytometer. eFluor 670-labeled iRBC were mixed with primary NK cells in the presence or absence of IgG for 1 h at 37°C. Granzyme B substrate was then added to the cells and eFluor670+ granzyme B+ iRBC were quantified by flow cytometry. Filled symbols represent independent experiments with NK cells from three individual donors. Open symbols represent samples incubated in the absence of IgG.

could be emitted from NK cells, NK–iRBC conjugates were dissociated with EDTA prior to addition of the granzyme reporter. Granzyme B activity was detected in iRBC after incubation with NK cells in the presence of IgG for 1 hour, and the signal increased at a higher NK to

iRBC ratio (Fig 2G). A benefit of this reporter assay is that it ignores spontaneous lysis of iRBC. Indeed, background signal in the absence of either NK cells or anti-RBC IgG was negligible (Fig 2G). This experiment provided strong evidence that iRBC are lysed by NK cells via polarized release of lytic effector molecules upon conjugate formation.

Experiments described so far gave quantitative information on the damage inflicted to iRBC and to intracellular parasite during ADCC by NK cells. However, they did not distinguish between a single, sequential pathway (iRBC damage preceding parasite damage) or two pathways, one leading to iRBC damage alone, and another leading to simultaneous damage to iRBC and parasite. Live imaging experiments were performed to examine single cells over time. Primary NK cells and iRBC were stained with eFluor 450 and eFluor 670, respectively, and co-incubated at a ratio of 3 iRBC to one NK cell. PI and anti-RBC IgG were present during the incubation and images were acquired for 5 hours at one frame per minute in a temperature-controlled chamber. Single frames from a movie (S1 Movie) show a motile NK cell encountering an iRBC and forming a tight synapse, followed 8 minutes later by a PI signal (Fig 3A). In this case, formation of a ghost cell could not be determined, as the NK cell covered the iRBC at 3 minutes.

The second example (Fig 3B) shows a slower event with well demarcated steps: an NK cell remained still for almost an hour after encounter with an iRBC before it established a committed synapse (at 55 minutes) and became activated (S2 Movie). The iRBC target turned into a ghost 25 minutes later, as detected by loss of eFluor 670 and became PI+ 18 minutes later (Fig 3B). NK cells mostly remained attached to damaged iRBC (Figs 3B and S3). The next example is a serial killer NK cell that damaged three independent *P.f.* parasites in different iRBC 16 minutes after the first contact with an iRBC, and 18 minutes and 36 minutes after contact with two more iRBC (S3 Fig, S3 Movie). That NK cell had been activated initially by contact with a ghost cell that harbored a PI-negative parasite. The PV in the ghost iRBC became PI+ 3 minutes after contact with the NK cell (S3 Fig). Our results clearly support the sequential pathway model of NK-dependent damage to iRBC preceding damage to *P.f.* within parasitophorous vacuoles. Free ghost iRBC (i.e. not in contact with an NK cell) housing a PI-negative PV for a long time (e.g. >67 minutes in S3 Fig, S3 Movie) were common, suggesting that NK cells accelerate the damage to *P.f.* inside ghost iRBC.

## Release of *P.f.* parasitophorous vacuoles from iRBC by NK cell-mediated ADCC

To obtain higher resolution images of NK–iRBC synapses and to examine in finer detail the damage caused by NK cells during ADCC, transmission electron microscopy (TEM) and scanning electron microscopy (SEM) were performed using fixed samples of NK cells that had been incubated with iRBC for 3 hours in the presence of 2 μg/ml anti-RBC IgG. By TEM, an apparently intact iRBC shows a late-stage parasite in a PV, including the characteristic knobs at the plasma membrane and hemozoin crystals in the digestive vacuole (Fig 4A). The second TEM image shows an iRBC ghost with a collapsed plasma membrane, which has retained knobs at its surface (Fig 4A). The integrity of the PV seems compromised. By contrast, uninfected RBC did not interact with NK cells even in the presence of IgG (S4 Fig). An example taken at a lower magnification shows a single, damaged iRBC engaged in two synapses with well polarized NK cells on either side (S4 Fig). One of the NK cells is also engaged in a second synapse with a visibly damaged iRBC.

Three SEM images were chosen as examples of NK–iRBC synapses with increasing evidence of iRBC damage (Fig 4B). In the first, the iRBC is still rigid and carries the characteristic knobs at the plasma membrane, despite a large surface area already engaged in a synapse with

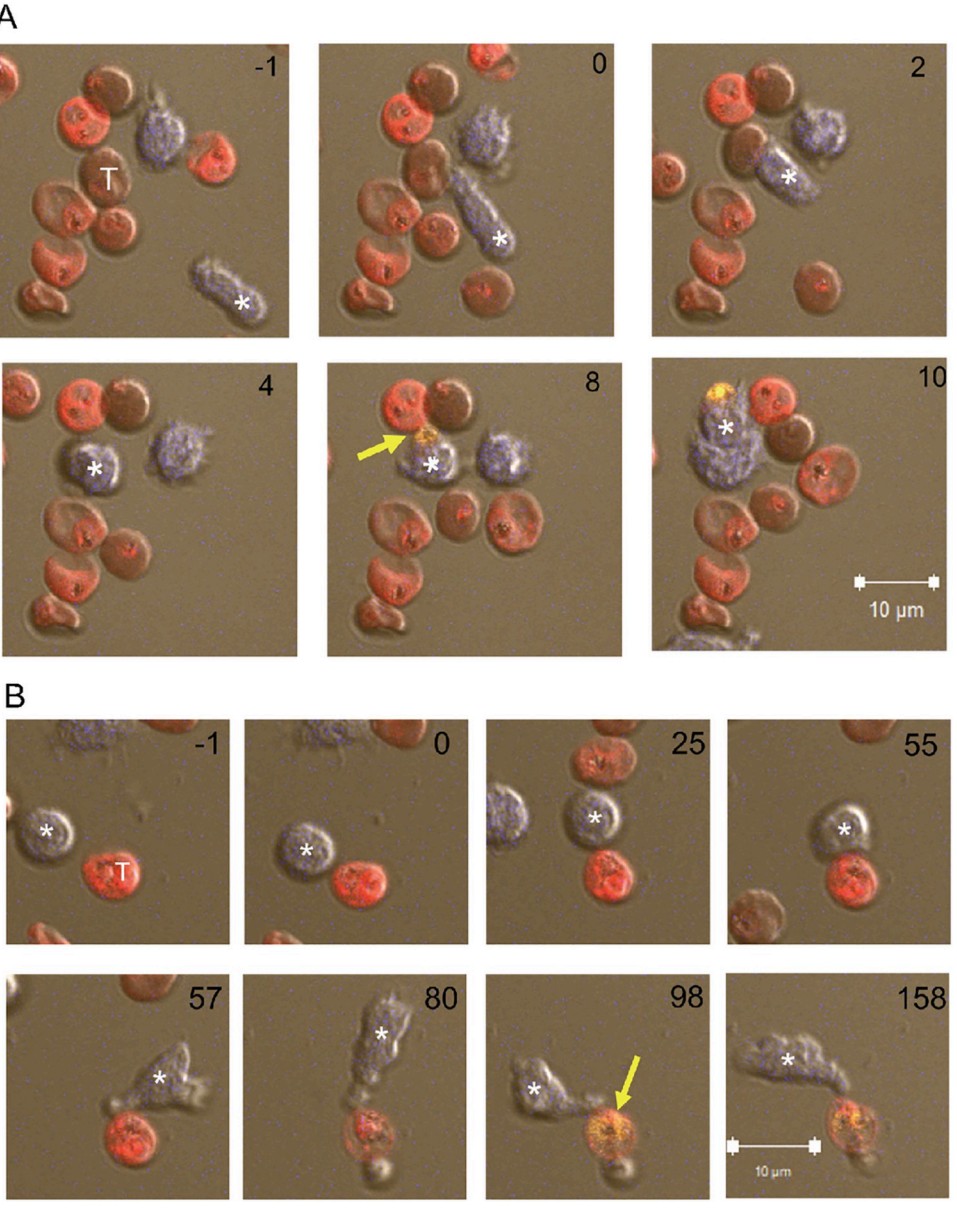

**Fig 3. Live imaging of iRBC incubated with NK cells. (A, B)** eFluor 450-stained primary NK cells (blue) incubated with eFluor 670-stained iRBC (red) at a ratio of 3:1 in the presence of 2 μg/ml anti-RBC IgG. PI was added at the start of image recording. The number in each panel indicates the time in minutes relative to the first NK–iRBC contact (set as minute 0). The effector NK cell is marked with an asterisk and the iRBC target with a T in the first frame. The first appearance of a PI-positive parasite is marked with a yellow arrow. Six time frames spanning 11 minutes are shown in (A) and eight timeframes spanning 159 minutes are shown in (B). The scale bar represents 10 μM.

an NK cell. In the next image, the synapse appears tighter and the iRBC has lost its rigidity. The third image reveals greater damage and visible perforation in the iRBC membrane, suggesting that it may be a ghost (Fig 4B). Additional examples of NK–iRBC synapses are shown in S4 Fig.

Comparison of low magnification SEM images of NK cells cocultured with iRBC was informative (Fig 4C). In the absence of anti-RBC IgG, NK cells and iRBC were evenly distributed, bar the occasional contact (Fig 4C, *left* and S4 Fig). By comparison, the culture imaged after

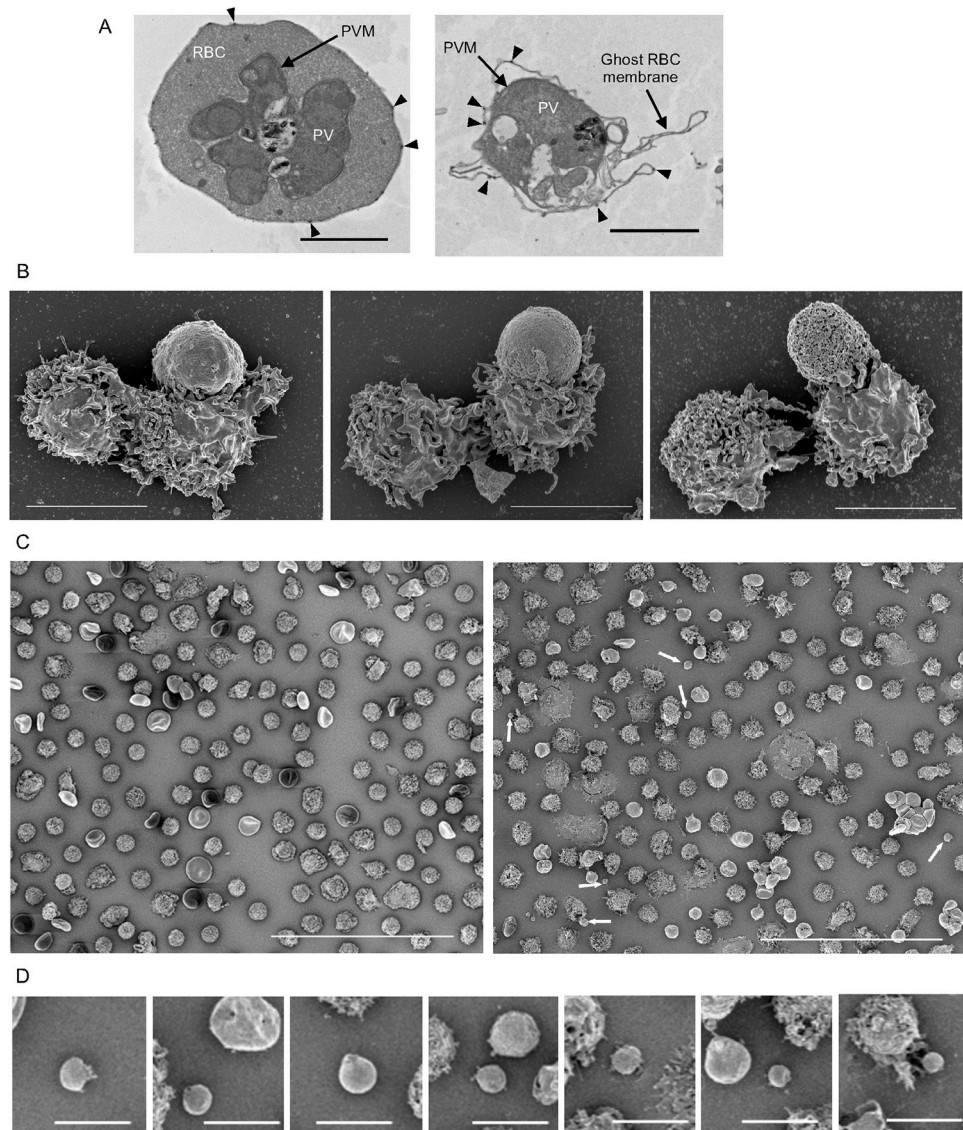

**Fig 4. Electron microscopy images of iRBC and NK cell cocultures.** (**A**) NK cells and iRBC were coincubated at a ratio of 5:1 in the presence of 2 μg/ml anti-RBC IgG for three hours. Samples were fixed and prepared for analysis by TEM. An apparently intact late stage trophozoite with well demarcated iRBC plasma membrane and parasitophorous vacuole membrane (PVM) is shown in the first panel. The second panel shows an iRBC ghost cell containing a PV. Characteristic knobs of the iRBC plasma membrane are visible on the intact and the ghost membranes (arrowheads). Scale bars are 2 μm. (**B**) NK–iRBC conjugates imaged by SEM after a coincubation as described in (A). Examples of NK–iRBC conjugates with increasing apparent damage to the iRBC membrane are shown from left to to right. In the first panel the iRBC in contact with an NK cell was still rigid and knobby, suggesting it is relatively intact. In contrast, a round and smooth iRBC (middle) with extensive contact with an NK cell indicates a loss of cytoskeleton structure and a round and apparently perforated iRBC membrane (right) suggests it was a ghost iRBC (right). Each panel includes a second NK cell, not involved in the contact with the iRBC. Scale bars are 5 μm. (**C**) A low magnification SEM image of NK cells and iRBC after co-incubation in the absence of IgG (left panel) shows NK cells and iRBC that appear mostly intact. A similar frame taken after coculture in the presence of IgG (right panel) is more heterogeneous and includes vesicles that are smaller than NK cells or iRBC (white arrows). The diameter of such vesicles, including any that was at least 1.5 μm wide, had a mean of 2.23 μm (SD 0.46 μm, n = 12). Scale bars are 50 μm. (**D**) A sample of vacuole-like particles seen after NK cell-mediated lysis of iRBC from two independent experiments. Frame 2 and 6 include a typical ghost iRBC. Scale bars are 5 μm.

NK-mediated ADCC looked very different (Fig 4C, *right* and S4 Fig). Damaged iRBC are abundant and some of the NK cells have collapsed. Notably, smaller vesicles ranging from 1.5 μm to 3.1 μm were visible (Fig 4C, *right* and 4D). Such vesicles were not detected in iRBC–NK co-cultured in the absence of anti-RBC IgG (Fig 4C, *left*). Zoomed-in images of these vesicles revealed a smoother surface than that of ghost iRBC, suggesting that they were free PV (Figs 4D and S4). The mean diameter of vesicles released from iRBC was 2.23 μm (n = 12, SD 0.46 μm). We conclude that damage inflicted to iRBC during NK-mediated ADCC results in formation of ghost cells and that some may have been sufficiently damaged for a PV to escape.

To confirm that particles released during coculture of NK cells and iRBC were *bona fide* PV that encapsulate developing *P.f.* parasites, and to follow their fate during live imaging, we generated a 3D7 *P.f.* strain in which the wild-type EXP2 gene was replaced with a gene encoding EXP2 tagged with GFP at the C-terminus. The exported protein-2 (EXP-2) is the pore-forming subunit of a protein translocon in the PV membrane. The EXP2 C-terminus is inside the vacuole [35]. Expression of EXP2-GFP indeed revealed PV with a green membrane in both intact iRBC and ghost cells (Fig 5A and 5B). The intact iRBC is eFluor 670-bright and phalloidin-negative, while the ghost cell is eFluor 670-dim phalloidin+ and distinctly translucent in the DIC image. An example of each is shown in a single image (Fig 5B). Live imaging was performed to examine the fate of PV during NK cell-mediated lysis of iRBC. EXP2-GFP+ iRBC were incubated with primary NK cells in the presence anti-RBC IgG and phalloidin. Frames from a movie spanning 12 minutes show 2 NK cells close to 3 iRBC (Fig 5C, S4 Movie). The first NK–iRBC contact (time 0) resulted in a phalloidin+ iRBC after 7 minutes. The next NK–iRBC contact (t = 2 min) resulted in a phalloidin+ iRBC 4 minutes later with a GFP+ PV visible inside the phalloidin+ ghost 2 minutes later (Fig 5C). The first NK–iRBC contact (which turned phalloidin+ after 7 minutes) released a free GFP+ PV after 9 minutes (Fig 5C). In the second example, two NK cells contact and cover one iRBC each (Fig 5D). The first contact led to a GFP+ PV inside a phalloidin+ iRBC 12 minutes later. Twelve minutes after the second NK–iRBC contact (t = 7min), a third NK cell contacts the iRBC still engaged with the second NK cell (t = 19 min). The iRBC became phalloidin+ 3 minutes later and a free GFP+ PV emerged another minute later (Fig 5D). We conclude that individual encounters of NK cells with iRBC during ADCC result in iRBC membrane damage, as determined with phalloidin, and that PV are either retained within or released from ghost cells.

To quantify the release of free PV during NK-mediated ADCC, iRBC stained with PKH26 and eFluor 670 were incubated with PKH67-stained NK cells for 1 hour and analyzed by flow cytometry. After gating out PKH67+ NK cells, PV were identified as PKH26-negative eFluor 670+ events (Figs 6A and S5). The fraction of eFluor 670+ events that were PKH26-negative increased with higher NK to iRBC ratios and with higher IgG concentrations (Fig 6A and 6B). These results showed that free PV observed in imaging experiments are not uncommon and that PV release from iRBC depends on NK cells and ADCC. Captured on a TEM image, a PV released during NK-mediated ADCC had a round shape and a smooth membrane (Fig 6C), consistent with SEM images (Figs 4D and S4). Therefore, lysis of iRBC by NK cells left sufficient GFP+ PV structurally intact to be detectable by quantitative flow assays and by imaging experiments with live or fixed samples.

## IgG in plasma of malaria-exposed individuals binds to free parasitophorous vacuoles

We developed a protocol to isolate and purify PV for further study. EXP2-GFP iRBC stained with PKH26 and eFluor 670 were treated briefly with 0.0125% saponin to gently permeabilize

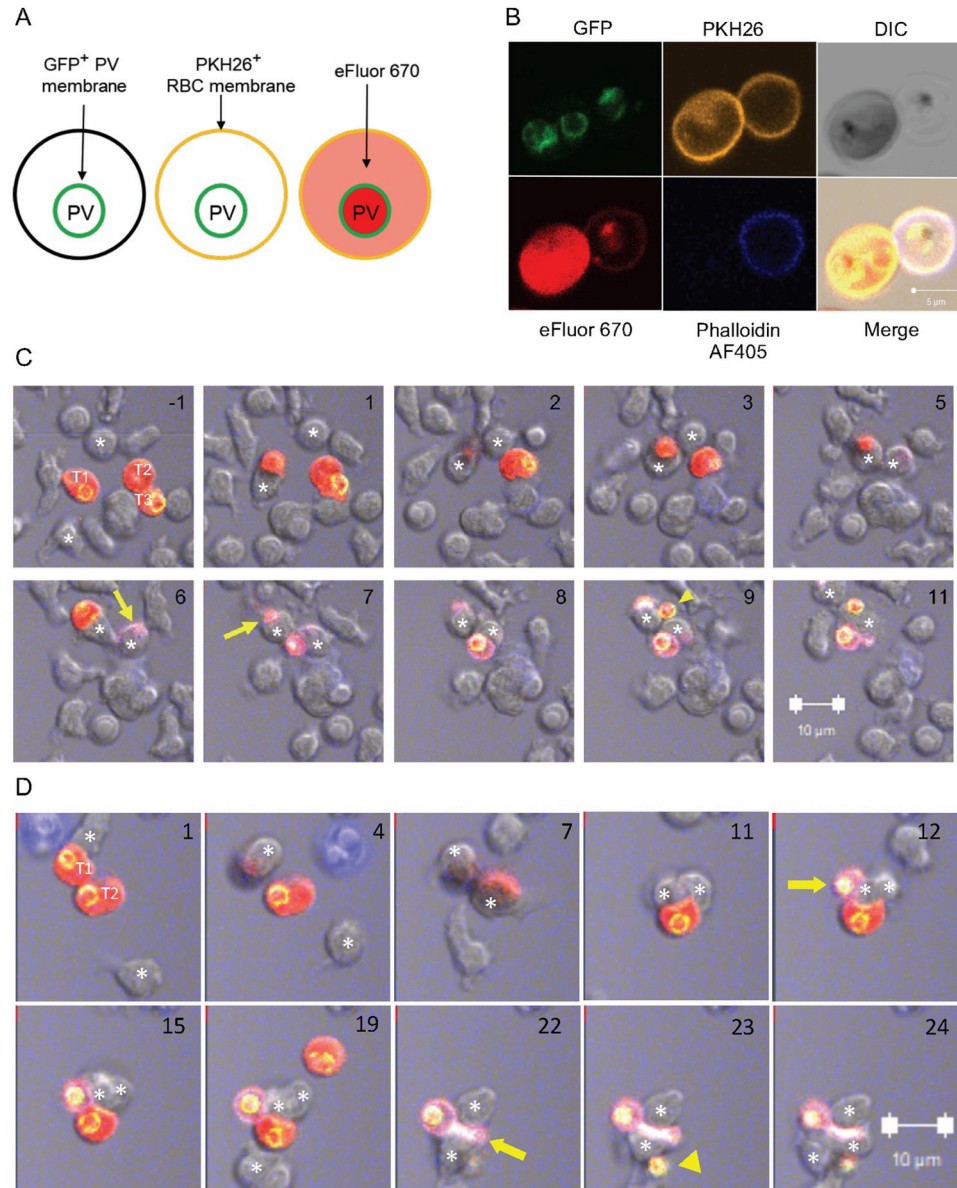

**Fig 5. Release of parasitophorous vacuoles from iRBC during incubation with NK cells.** (**A**) Diagram of a strategy to identify PV that are either inside or outside iRBC. 3D7 *P.f.* expressing PfEXP2 tagged with GFP was used to track PV. EXP2-GFP+ iRBC were stained with membrane dye PKH26 and with eFluor 670. (**B**) Immunofluorescence image of EXP2-GFP+ iRBC pre-stained as in (A) before incubation with NK cells. The cell on the right is a ghost, as seen by Phalloidin AF405 staining and in the DIC image. The scale bar is 5 μm. (**C**) Ten frames spanning 12 minutes. Unstained NK cells and dual-stained EXP2-GFP+ iRBC were incubated with 2 μg/ml anti-RBC IgG. Two effector NK cells are marked with an asterisk in every frame, and three iRBC targets are marked T1, T2, and T3 in the first frame. The first contact of the NK cell at the bottom with T1 was set as time 0. The NK cell at the top contacted T2 2 minutes later. The first phalloidin+ iRBC (yellow arrow) was detected 4 minutes after the first NK–iRBC contact. T1 becomes phalloidin+ 7 minutes after contact (yellow arrow). Due to cell overlaps, it was not possible to unambiguously follow the respective fate of targets T2 and T3. The first GFP+ PV visible inside a phalloidin+ ghost iRBC appeared after 8 minutes. After 9 minutes, a second GFP+ PV was detected (yellow arrowhead), which was apparently outside of an iRBC. (**D**) Ten frames spanning 23 minutes from an experiment performed as in (C). The first NK–iRBC (T1) contact was visible at time 0, as shown at the top of the image (taken 1 minute later). The NK cell covered the iRBC at 4 minutes. A second effector NK cell moved from the bottom towards T2 and covered it at 7 minutes. The first visible GFP+ PV inside a ghost (yellow arrow) was detected after 12 minutes. At 19 minutes, a third effector NK cell (asterisk at the bottom) contacted the other, still intact iRBC, which became phalloidin+ 3 minutes later (yellow arrow). A free GFP+ PV emerged one minute later (yellow arrowhead). The respective positions of the 3 effector NK cells and 3 iRBC could not be tracked unambiguously. (C and D) For clarity, the yellow PKH26 signal was removed from images. Numbers in the top right corner indicate time in minutes. Scale bars are 10 μm.

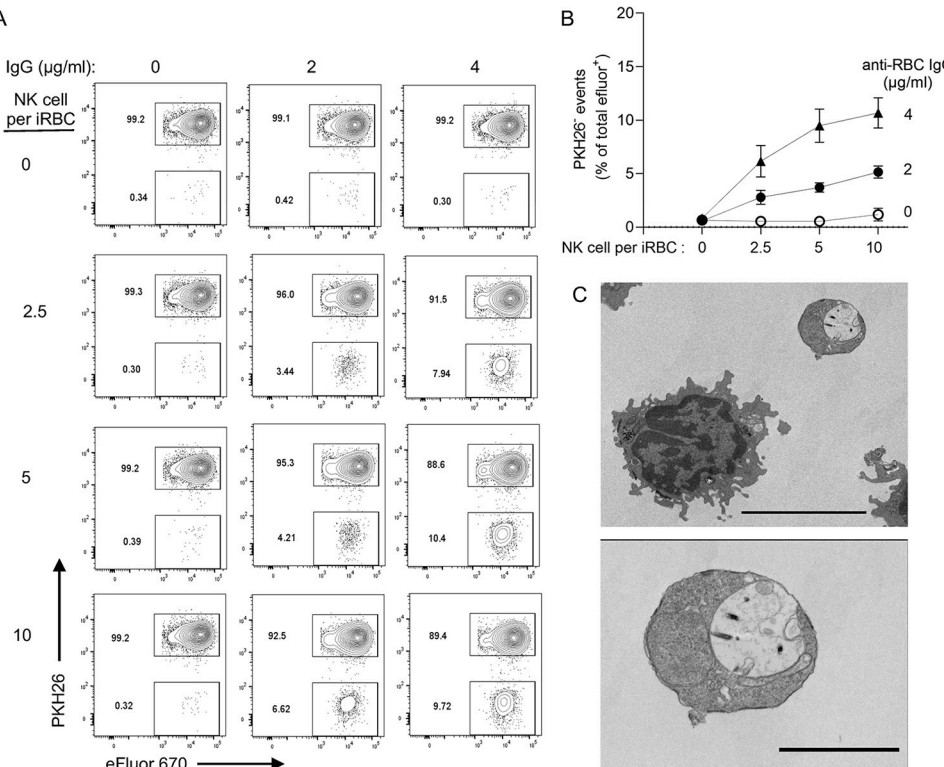

**Fig 6. Quantitative analysis of PV release from iRBC during incubation with NK cells.** (**A**) Flow cytometry analysis to detect PKH26-negative eFluor+ particles (i.e. free PV) after incubation with NK cells for one hour at the indicated IgG concentrations and NK cell to iRBC ratios. (**B**) PKH26-negative eFluor+ particles increased with higher IgG concentration and NK to iRBC ratios. Error bars represent standard error of the mean from four independent experiments. (**C**) TEM image of a free PV (top right) and an NK cell taken from an experiment similar to those in (A) and (B). Scale bar is 5 μm. The PV is shown at a higher magnification in the lower image with a scale bar representing 2 μm.

the plasma membrane. After a wash in PBS, the iRBC plasma membrane was further disrupted by mechanical force through a needle (Figs 7A and S6). Samples were sorted by flow cytometry for eFluor 670+ PKH26-negative particles (Figs 7A and S6). Confocal microscopy showed the expected small vesicles with a GFP+ membrane and a pigmented appearance in the DIC image (Fig 7A). SEM images of purified PV (Fig 7B) showed round shapes in the same size range as vesicles released after incubation of iRBC with NK cells (Figs 4D and S4).

We then examined the reactivity of isolated PV with antibodies, including polyclonal anti-RBC IgG and Abs in plasma of adults exposed to seasonal *P.f.* infection. Anti-RBC IgG, which stained uninfected RBC and iRBC equally well (S1 Fig), had very low binding to PV (Figs 7C and S6). Two conclusions can be drawn: first, the PV preparation was not contaminated by particles or vesicles derived from the plasma membrane of iRBC, and second, the PV membrane is antigenically distinct from RBC membrane. Next, we tested binding of IgG from plasma of malaria-exposed adults from a cohort in Mali (referred to as Mali IgG). Mali IgG at 0.8 mg/mL bound to iRBC but not to uninfected RBC (Fig 7C and 7D). Binding to iRBC was low but significant, as IgG from healthy US donors did not bind. In contrast, the MFI of Mali IgG bound to PV was two orders of magnitude higher than the MFI of control US IgG (Fig 7C and 7D). Neither IgG bound to uninfected RBC (Fig 7C and 7D). Therefore, malaria-exposed adults have a high titer of antibodies elicited against PV released from lysed iRBC.

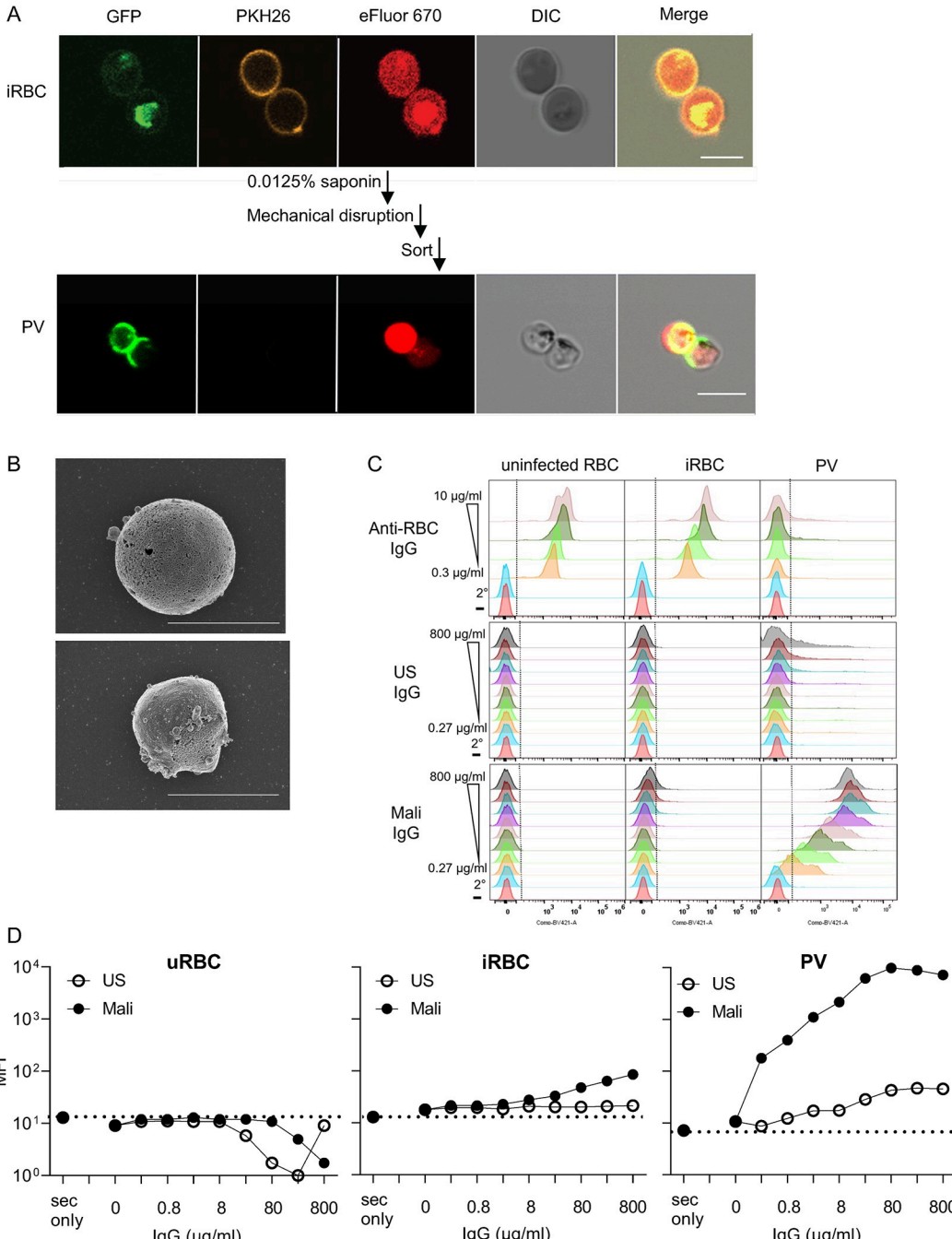

**Fig 7. IgG in plasma of malaria-exposed adults binds PV.** (**A**) Immunofluorescence images of intact EXP2-GFP+ iRBC (top panels) and PV isolated from iRBC by mild saponin treatment and mechanical disruption of iRBC membrane, followed by flow cytometry sorting of PKH26-negative eFluor+ particles (bottom). Scale bars are 5 μm. (**B**) SEM images of PV isolated as in (A). Their diameter is 3.19 μm (*top*) and 2.56 μm (*bottom*). Scale bar is 3 μm. (**C**) Profiles of uninfected RBC, iRBC, and PV, as indicated, stained with anti-RBC IgG (top), IgG from healthy US donors (US, middle) and IgG from malaria-exposed adults in Mali (Mali, bottom). Human IgG samples were titrated over a 3000-fold range, as indicated. The bottom two rows in each panel show staining with the secondary antibody alone (2°) or in the absence of IgG (–). (**D**) Mean fluorescence intensity (MFI) of uninfected RBC (uninfected RBC), iRBC, and PV stained as in (C) with US IgG (open circles) and Mali IgG (filled circles).

## Mali IgG induces clearance of PV by monocytes

IgG bound to free PV could have the benefit of stimulating clearance by monocytes through phagocytosis. To test this, purified eFluor 670+ PV were incubated with primary monocytes for one hour and analyzed by flow cytometry for the presence of eFluor+ monocytes (Fig 8A). The proportion of eFluor+ monocytes in the presence of Mali IgG was 3-fold greater than it was in the presence of US IgG or in the absence of antibody. To exclude the possibility that the eFluor 670 signal detected in the monocyte gate was due to an extracellular association with PV rather than phagocytosis, we used a pHrodo dye that emits fluorescence at an acidic pH. Thus, a positive signal in monocytes would indicate not only phagocytosis but also delivery of PV into acidic phagosomes. The dual stained iRBC were further stained with pHrodo and then incubated with monocytes in presence or absence of different antibodies. Under the same conditions as with eFluor+ PV, the proportion of pHrodo+ monocytes detected after incubation with PV in the presence of Mali IgG was 5-fold greater than it was in the presence of US IgG or in the absence of antibody (Figs 8A and S7). The greater selectivity for PV phagocytosis measured by pHrodo was due mostly to lower values obtained in the absence of Mali IgG. We conclude that our assay was measuring *bona fide* phagocytosis of PV by monocytes.

To rule out a contribution to PV phagocytosis by some artifact introduced during isolation of PV from iRBC, we tested phagocytosis of PV that had been released from iRBC during NK-cell mediated ADCC and isolated by flow cytometry sorting. Six independent experiments, each using primary monocytes and NK cells obtained from different malaria-naïve autologous donors, and fresh trophozoite-enriched iRBC cultures, showed that phagocytosis of PV that had been released from iRBC co-cultured with NK cells was as good as those PV prepared from saponin-treated iRBC (Fig 8B). Finally, to directly observe PV phagocytosis, we performed live imaging of monocytes incubated with eFluor+ PV in the presence of Mali IgG. Imaging revealed rapid internalization of PV by monocytes (Fig 8C), thus validating our measurements in quantitative flow cytometry assays.

## Discussion

The activity of human NK cells, particularly adaptive NK cells [36] and CD56-negative NK cells, has been associated with resistance to malaria [26,27]. How they provide protection is still an open question. To address this, we studied the interaction between primary NK cells and erythrocytes infected with *P. falciparum* (iRBC). We show that stable adhesion between the two was dependent on two pairs of receptor–ligand interactions and that damage inflicted to iRBC by NK cells required activation of FcR CD16 by IgG bound to iRBC. These requirements are similar to the steps involved in lysis of tumor cells or virus-infected cells by NK cells except for the receptors involved. Instead of a combination of tumor cell ligands or stress signals that coengage innate NK receptors [15], NK cells responding to iRBC were activated by IgG, which is sufficient to trigger CD16 and induce degranulation to release lytic effector molecules [16]. While NK cell adhesion to tumor cells is mediated by integrin αLβ2, NK cells use integrin αMβ2 and receptor CD2 to interact with ICAM-4 and CD58 on iRBC, respectively. Clear evidence of these two receptor–ligand interactions for adhesion between NK cells and iRBC was obtained using differentiated erythrocytic cells carrying specific deletions of the genes *ICAM4* and *CD58*. A fundamental difference between ICAM-4 and ICAM-1 is the absence of a glutamic acid (Glu34 in ICAM-1) in ICAM-4 that is crucial for strong ICAM-1 binding to αLβ2 via coordination with $Mg^{2+}$ [37]. A weaker ICAM-4–αMβ2 interaction could be why adhesion of NK cells to iRBC required inside-out signaling by co-engagement of CD16 by IgG bound to iRBC.

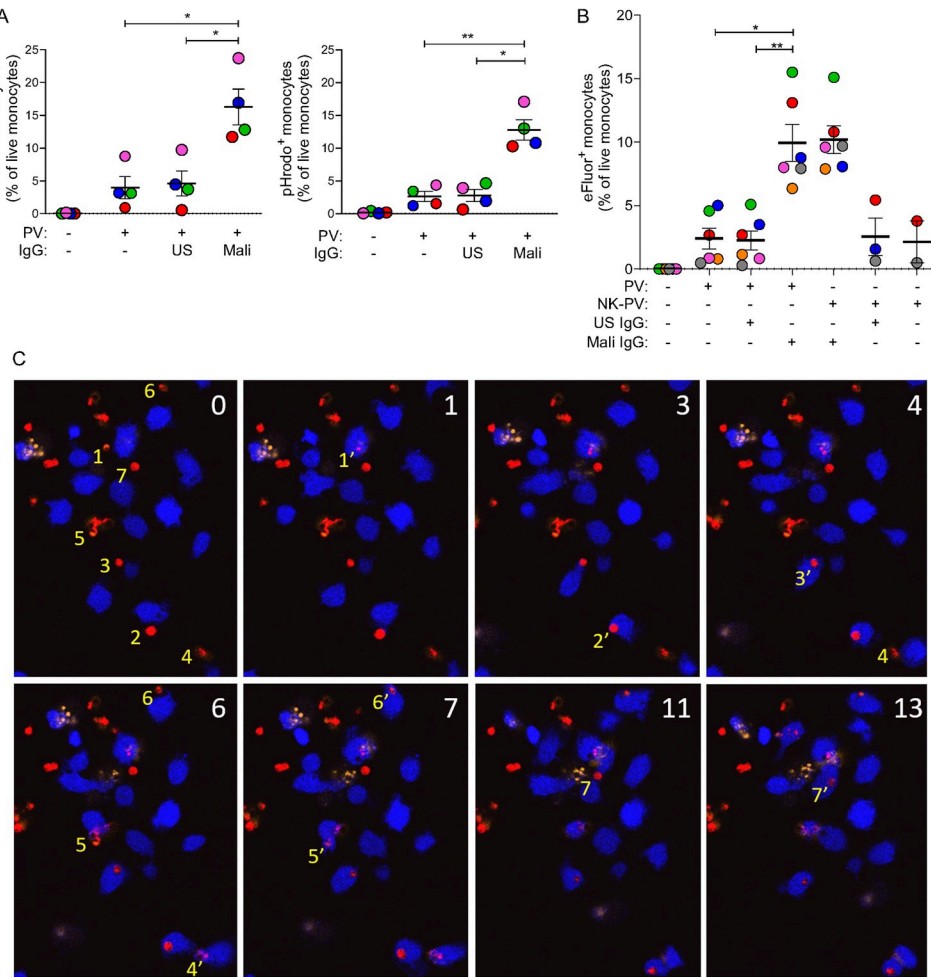

**Fig 8. Phagocytosis of PV by primary monocytes in the presence of IgG from malaria-exposed adults.** (**A**) eFluor 670+ (left panel) and pHrodo+ (right panel) PV isolated from iRBC, as described in Fig 7, were incubated for one hour with primary monocytes in the absence or presence of 80 µg/ml US IgG, and 80 µg/ml Mali IgG. Phagocytosis was determined by quantifying the proportion of eFluor 670+ or pHrodo+ monocytes by flow cytometry. Error bars represent standard error of the mean. Four independent experiments were performed with monocytes from different donors, identified by different colors, and with independent PV preparations. (**B**) Experiments performed with PV isolated from iRBC (PV) as described in Fig 7, or PV isolated after incubation of iRBC with NK cells (NK-PV) at a NK to iRBC ratio of 6:1 for 3 hours. Due to limitations in the number of PV released and isolated after NK-mediated lysis of iRBC, the no Ab control was performed in 2 out of 6 experiments and the US IgG control in 3 out of 6 experiments. The crucial comparison of US versus Mali IgG was performed in all experiments. Error bars represent standard error of the mean from six independent experiments. (**C**) Live imaging of primary monocytes stained with eFluor 450 (blue) and co-incubated at a ratio of 1:1 with PKH26-negative eFluor 670+ PV (red) in the presence of 80 µg/ml Mali IgG. Numbers in the top right corner of each image indicate time in minutes. PI was added at the start of image acquisition to exclude damaged monocytes and PV. At time 0, seven PV that were phagocytosed within the next 13 minutes are numbered (yellow) to their left in the order of subsequent phagocytosis by a monocyte. For example, PV1 at time 0, which was phagocytosed after one minute, is labeled 1'. Subsequent phagocytosis events are labeled 2', 3', to 7'. The eFluor 670 signal of PV after phagocytosis diminished rapidly, except for PV2, still visible 10 minutes later. PV2 may represent a PV retained in a phagocytic cup prior to complete endocytosis. * p < 0.05, ** p < 0.01 Sidak multiple comparisons test.

We found that uninfected RBC did not form conjugates with NK cells, despite the presence of opsonizing IgG (Fig 1A–1C). A possible reason is the high deformability of erythrocytes, which is required for their passage through the spleen [38]. *P.f.* invades deformable erythrocytes and rigidifies its host membrane in order to adhere to vessels for sequestration [39,40]. It

is likely that this iRBC rigidity also facilitates adhesion to effector cells of the immune system [41]. In addition to its role in adhesion, CD2, the receptor for CD58, is also an amplifier of signaling by CD16 on NK cells and by the T cell receptor [24,25,42]. CD2 expression is upregulated in adaptive NK cells [36]. It is therefore likely that CD2 on NK cells contributes not only to adhesion but also to the ADCC response to infected erythrocytes.

We used several assays to determine the integrity of iRBC and of *P.f.* in parasitophorous vacuoles following incubation with IgG and NK cells. Imaging and quantitative flow cytometry were used as complementary approaches to show that NK cells are activated after contact with iRBC in the presence of IgG, as seen by increased motility and frequent contacts with other iRBC. Unlike γδ2 T cells, which detect and phagocytose iRBC by binding to butyrophilin on iRBC [43], we did not observe phagocytosis of opsonized iRBC by NK cells. Serial killing by NK cells of *P.f.* inside PV within iRBC was observed. This property would render NK cells very effective at sites of iRBC sequestration in the microvasculature of organs such as the lungs, spleen, and bone marrow [44].

NK cell-dependent ADCC does not destroy iRBC and PV but permeabilizes and damages the iRBC plasma membrane. Two reliable readouts of iRBC damage were the release of hemoglobin and the accessibility of F-actin in iRBC to phalloidin. The PV membrane remained structurally intact even for those PV that were released from ghost iRBC after incubation with NK cells. However, damage to the parasite occured, as shown by accessibility of DNA to propidium iodide. In addition, active granzyme B was detected in iRBC during NK-mediated ADCC, much like GzmB delivered into tumor target cells. Granzyme cleaves *P.f.* proteins at the intra-erythrocytic blood stage, as was shown with γδ2 T cells [45]. Permeabilization of iRBC requires granulysin rather than perforin due to cholesterol depletion from the iRBC plasma membrane at late stages of *Plasmodium* development [43,45,46].

Malaria is an inflammatory disease and suppression of the innate host immune response contributes to protection from malaria symtpoms [47]. AT-rich *P.f.* DNA activates a DNA-sensing pathway mediated by STING and IRF3/IRF7 [48]. Hemozoin released from ruptured iRBC during egress of merozoites is associated with *P.f.* DNA and activates TLR9 and the NLRP3 inflammasome [49,50]. Hemozoin-induced responses cause dendritic cell dysfunction, which impairs anti-malarial immunity [51]. Thus, while early innate proinflammatory responses by myeloid cells control parasite replication and promote parasite clearance, elevated levels are tied to severe malaria. An indication that dampening of myeloid cell responses is critical for disease tolerance comes from evidence that *P.f.* induces epigenetic reprogramming of myeloid cells toward a regulatory state [9], a phenotype that associated with resistance to malaria [12].

*P.f.* at the blood stage depends on erythrocytes into which an invading merozoite begins a developmental cycle from ring to trophozoite, and to schizont until the RBC and PV membranes rupture to release fresh merozoites. The parasite remodels erythrocytes for its own purpose, modifies the plasma membrane lipid composition and inserts its own ion channel and variant antigens that serve to take up nutrients and as ligands to mediate sequestration. Therefore, structural integrity of the iRBC membrane is required for *P.f.* development. Nevertheless, the fate of a damaged parasite inside a PV still matters because release of parasite DNA and components of the digestive vacuole induce inflammation via TLR9 and type I interferon, and disable myeloid cells [47,49,51].

We found that isolated PV, free of iRBC plasma membrane, reacted strongly with IgG from plasma of malaria-exposed adults living in Mali, most of whom had acquired clinical immunity to malaria. Therefore, antigens exposed on the PV membrane after release from iRBC must induce an Ab response. During invasion by a merozoite the PV is formed by invagination of the erythrocyte plasma membrane [52], a process that includes selective sorting of proteins

and lipids [53,54]. For example, stomatin is excluded from the PV while flotillin is included despite their common presence in detergent-resistant membrane domains of erythrocytes [55]. Likewise, phosphatidylserine (PS) is included while phosphatidylinositol [4,5] diphosphate (PIP2) is not [53]. CD58 is among proteins that are included in the PV membrane during merozoite invasion. Note, however, that the inner leaflet of the PVM represents what was facing out at the RBC plasma membrane. Therefore, the CD2-binding domain of CD58 is expected to be *inside* the PV [55]. Although ligands for human IgG on PVM are not known, anything not normally exposed to circulating blood may induce an Ab response, and that includes cytosolic domains of human proteins internalized during *P.f.* invasion and *P.f.* proteins that span the PVM, such as EXP2. PV released from iRBC are clearly immunogenic, as the titer of PV-binding IgG from malaria-exposed individuals is far higher than IgG that binds to iRBC. This IgG induces efficient phagocytosis of PV by monocytes. Thus, clearance of PV before further damage and release of parasite material may help control inflammation associated with *P.f.* infection.

NK cell ADCC activity in people exposed to malaria is associated with reduced parasitemia [26,27]. Adaptive NK cells have a higher intrinsic ADCC response [25,26,36]. Remarkably, in the context of malaria, adaptive NK cells triggered by CD16 upregulate degranulation but not IFN-γ secretion [10,26], perhaps to constrain inflammation. The type I interferon response that occurs during malaria [47,48] may contribute to the control of inflammation by inhibiting the production of interferon-γ by NK cells [56]. A study of peripheral blood mononuclear cells in individuals living in a region of low malaria transmission found that NK cells during a malaria episode upregulated effectors of cytotoxicity (*GNLY*, *FCGR3A*, *GZMB*) but also markers of suppression, such as co-inhibitory receptors *HAVCR2* (Tim-3), *LAG3* (CD223), and several *TNFRSF* members [10].

In summary, we show that NK cells recognize *P.f.* infected erythrocytes and adhere to them via CD2–CD58 and αMβ2–ICAM-4 interactions. This adhesion event is followed by damage to the iRBC plasma membrane and the release of parasitophorous vacuoles. These PV show robust binding to IgG from malaria-exposed individuals from endemic areas and this marks the PV for clearance by monocytes through phagocytosis. These findings provide insights into earlier observations that NK cell frequency and function correlated with decreased parasitemia and resistance to malaria [26,27] and reveal parasite clearance from the blood by NK-mediated ADCC in cooperation with monocytes.

## Materials and methods

### Ethics statement

Peripheral blood samples from healthy US adults were obtained from the NIH Department of Transfusion Medicine under an NIH Institutional Review Board-approved protocol (99-CC-0168) with written informed consent. The Mali cohort study was approved by the Ethics Committee of the Faculty of Medicine, Pharmacy, and Dentistry at the University of Sciences, Techniques, and Technologies of Bamako, in Mali and by the Institutional Review Board of the National Institute of Allergy and Infectious Diseases, National Institutes of Health. Prior to inclusion in this study, written informed consent was received from participants.

### Human antibody sources

Plasma samples were collected from adults enrolled in a multi-year malaria study in the rural village of Kambila, Mali [57], by starting with venous blood collected in citrate-containing cell-preparation tubes (Becton Dickinson). Samples were transported 45 km away to the Malaria Research and Training Centre in Bamako, where peripheral blood mononuclear cells (PBMC)

and plasma were isolated. Plasma was frozen in 1 mL aliquots at 80˚C. Samples were shipped to the United States on dry ice for analysis. Control serum from US individuals was obtained from Valley Biomedical.

## Enrichment of antibody

IgG was purified from plasma or serum by standard affinity chromatography. Briefly, each sample was diluted 1:2 in column equilibration-wash buffer (20 mM NaPO4, pH 7.0). The IgG fraction was isolated on Protein G Sepharose (GE Healthcare 17-0618-02) placed in columns (Bio-Rad, 732–1010), eluted with 100 mM glycine, pH 2.7 and immediately neutralized to pH 7.4 with 1 M Tris pH 8.0. IgG was concentrated and dialyzed in Amicon Ultra-15 centrifugal filter device (3 kDa MW cutoff) with PBS.

## NK cell isolation

Human blood samples from healthy US donors were drawn for research purposes at the NIH Blood Bank under an NIH IRB approved protocol with informed consent. PBMC were first isolated using Lymphocyte Separation Medium (MP Biomedical), washed with PBS twice, and resuspended in PBS, with 2% FBS and 1 mM EDTA. NK cells were isolated from PBMC by depletion of non-NK cells using an NK cell isolation kit (STEMCELL Technologies). The manufacturer's protocol was modified as follows. PBMC were resuspended at $2\times10^8$ cells/mL and 50 μL/mL of the cocktail was added. Anti-CD3 biotin (STEMCELL Technologies) was also added at a concentration of 2.5 μL/mL to increase NK cell purity. Resting NK cells were resuspended in Iscove's modified Dulbecco's medium (IMDM; Invitrogen) supplemented with 10% human serum (Valley Biomedical) and were used within 1 to 4 days after isolation. Only NK cell preparations that had greater than 95% CD14-, CD3-, CD56+ cells, as determined by flow cytometry, were used in experiments.

## Monocyte isolation

Monocytes were isolated by depletion of non-monocytic cells using a human monocyte enrichment kit without CD16 depletion (STEMCELL Technologies). PBMC were resuspended at $5\times10^7$ cells/mL and 50 μL/mL of the non-monocyte cocktail was added. Enriched cells were resuspended at $5\times10^7$ cells/mL in PBS with 2% FBS and 1 mM EDTA, and residual natural killer cells were depleted using 100 μL/mL of the CD56 Positive Selection Cocktail (STEMCELL Technologies). Monocytes ($5\times10^6$/vial) were cryopreserved in 10% DMSO in FBS. Only monocyte preparations with >95% CD14+ and <5% CD56+ cells, as determined by flow cytometry, were used for experiments.

## Cultivation and purification of *P. falciparum*

3D7 and 3D7-EXP2-GFP strains were cultivated at 37˚C under 5% $O_2$, 5% $CO_2$, 90% $N_2$ at 37˚C at <5% hematocrit using O+ human erythrocytes (Interstate Blood Bank, Inc.). Parasites were cultured in RPMI 1640 with glutamine, 25 mM HEPES, 0.5% Albumax, 0.26% sodium bicarbonate, 10 μg/mL gentamycin, and 50 mg/L hypoxanthine. Parasite development was monitored by light microscopy using methanol-fixed, Giemsa-stained thin blood smears. Parasites were synchronized using 5% sorbitol as described [6]. Parasite-iRBC were enriched for knobs using Zeptogel sedimentation routinely. For experiments, as described [6,58] infected RBC were enriched at the early trophozoite stage with Percoll-sorbitol gradient centrifugation, washed, and resuspended in complete medium.

## EJ cell maintenance and differentiation

EJ cells were maintained and differentiated as described previously [58]. Briefly, EJ cells were cultured in pIMDM medium containing Iscove's modified Dulbecco's medium (IMDM, Sigma-Aldrich) supplemented with 4 mM L-glutamine (Sigma-Aldrich), 330 µg/mL iron-saturated human holo-transferrin (BBI solutions), 10 µg/mL recombinant human insulin (Sigma-Aldrich), 2 IU/mL heparin sodium salt (Affymetrix), and 0.5% Pen/Strep (Life Technologies). Maintenance media was comprised of pIMDM supplemented with 5% solvent/detergent virus-inactivated plasma (octaplasma; Octapharma), 3 IU/mL erythropoietin (EPO; Roche), 50 ng/mL stem cell factor (SCF; R&D systems), 1 µg/mL doxycycline (Sigma-Aldrich), and 1 mM dexamethasone (Sigma-Aldrich). EJ cells were maintained at a concentration of $5 \times 10^4$–$7.5 \times 10^5$ cells/mL by splitting every 3 days. To induce differentiation, EJ cells were seeded at $2 \times 10^5$ cells/mL and cultured for 3 days in pIMDM supplemented with 5% octaplasma, 3 IU/mL EPO, 10 ng/mL SCF and 0.5 mg/mL doxycycline. On day 3, cells were seeded at $2 \times 10^6$ cells/mL on MS-5 stromal cells in pIMDM media containing 5% octaplasma and 3 IU/mL EPO. Cells were maintained in this media for the remainder of differentiation. On day 5 cells were reseeded at $5 \times 10^6$ cells/mL on fresh MS-5 stroma. Cells were harvested on day 8 of differentiation and layered on a 40% Percoll-PBS gradient. The pellet, which contained the terminally differentiated cells was washed thrice before use.

## EJ cell KO generation and validation

CRISPR/Cas9-mediated genetic perturbations of the EJ cell line were generated via adaptation of previously published protocols as described previously [58]. The LentiCas9-Blast plasmid (Addgene plasmid #52962) was first expressed in EJ cells via lentiviral transduction. Lentivirus was produced in HEK 293T using psPAX2 (Addgene plasmid # 12260) and VSV-G (Addgene plasmid # 12259) packaging plasmids. One million EJ cells were transduced with 1 mL of LentiCas9-Blast lentivirus via spinfection for 2 hours at 1000×g in the presence of 4 µg/mL polybrene. The transduced cells were then selected on 10 µg/mL blasticidin for 2–4 weeks. Next the LentiGuide-Puro plasmid [59] was used to express a single-guide RNA (CD58 sgRNA; 5′–GAGCATTACAACAGCCATCG–3′ and ICAM4 sgRNA: 5′–CCGGGAACACCTGCGTCACG–3′) LentiGuide-Puro (Addgene plasmid # 52963) was a gift from Feng Zhang. Lentivirus generation and transduction were performed as described above. After ~2 weeks of selection in 2 µg/mL puromycin, cells expressing the transgene were cloned via limiting dilution, and clones were screened for indels by PCR amplification. (CD58 primers: F 5′–GCTCAAGGAGTTTGTCTGCTCATC–3′ and R 5′–TGCTTGGGATACAGGTTGTC–3′; ICAM4 primers: F 5′–GCCTACAGTGAGGGACAGG–3′ and R 5′–ATCACGGGCTGCCAGAAG–3′) followed via Sanger sequencing. Editing was estimated using the Inference of CRISPR Edits tool (ICE, Synthego) as well as by flow cytometry by staining for ICAM4 and CD58.

## Conjugate assay

The assay was adapted from [28]. Resting NK cells were labeled with 1 µg/mL Cell Tracker Green CMFDA (Invitrogen) for 30 min at 37°C, washed and and resuspended at $1.5 \times 10^6$/mL. RBC or EJ cells were labeled with eFluor 670 (Molecular Probes) for 10 min at 37°C and washed. Labeled RBC or EJ cells were then incubated with 2 µg/mL of anti-RBC rabbit IgG (Antibody 209–4139, Thermo Fisher) in PBS for 30 min, washed in PBS and resuspended at $4.5 \times 10^6$/mL. NK cells (0.1 mL) and RBC or EJ (0.1 mL) were mixed at a 1:3 effector to target (E:T) cell ratio at 4°C, briefly spun down at 20 *g* for 3 min, and incubated in a 37°C water bath for 0, 20 or 40 min. Conjugate formation was stopped by vortexing and fixation of cells using 0.5% paraformaldehyde solution (Electron Microscopy Sciences). Conjugate formation was

determined by flow cytometry (FACScan, Becton Dickinson) and is represented as the fraction of NK cells that shifted into two color conjugates.

## Hemoglobin release assay

Resting NK cells were resuspended in CTS OpTmizer medium (GIBCO Invitrogen) at $5\times10^6$ cells/mL and 0.1 mL was plated in 96-well round bottom plates. Antibodies to NK cell surface receptors were added to cells at a final concentration of 10 μg/mL. EJ Cas9 control cells or EJ KO cells were layered on a Percoll gradient and centrifuged to ensure differentiated cells and resuspended at $2.5\times10^6$/mL. EJ cells were then incubated with anti-RBC rabbit IgG (1 μg/mL) in PBS for 30 min, washed and 0.1 mL was added to the NK cells. After 1 h at 37˚C, supernatants from the cocultures were tested in a hemoglobin release assay at a dilution of 1:1600 (Human Hemoglobin ELISA kit, Abcam) according to manufacturer's instructions. The following blocking antibodies to human CD molecules were used in this assay: CD2, clone TS1/8 (BioLegend, # 309236); CD11a, clone TS1/221 (Invitrogen, #MA11A10); CD11b, clone M1/70 (BioLegend, #101284); CD18, clone TS1/18 (BioLegend, #302116); CD49d, clone HP2/1 (Bio-Rad #MCA697).

## Granzyme assay

3D7-infected RBC targets were labelled with eFluor 670 and coated with anti-RBC rabbit IgG for 30 min. Target cells ($0.1\times10^6$) were mixed with resting NK cells at E/T ratios of 5 and 2.5 in 96-well V bottom plates for 1 h at 37˚C. Cells were then incubated with 5mM EDTA (30 min. at 4˚C) to disrupt NK–iRBC conjugates. Any residual conjugates were removed using forward scatter gating during flow cytometry. Granzyme B substrate (GranToxiLux kit, OncoImmunin Inc.) was added (5 min. at RT) and cells were acquired with a FACScan flow cytometer (BD BioSciences). The percentage of double positive eFluor+ and granzyme B+ target cells was calculated.

## Electron microscopy (EM)

Uninfected RBC and enriched iRBC ($2\times10^5$) either alone or mixed with resting NK cells ($1\times10^6$) were added to wells of a 96-well plate, in the presence of antibodies (either US IgG at 1.6 mg/mL, Mali IgG at 1.6 mg/mL, or anti-RBC rabbit antibody at 2 μg/mL) and incubated at 37˚C for 3 h. The cells were washed with 0.1 M phosphate buffer (pH 7.4), centrifuged at 2500 rpm for 5 min and supernatants were removed. The cells were again washed with 0.1 M phosphate buffer (pH 7.4) and resuspended in 20 μl of PBS. 2–3 drops of cells were dropped on the shiny side of polylysine coated Silicon chips (for scanning EM) and on polylysine coated plastic cover slips (for transmission EM). The cells were allowed to settle for 5 mins. The chips and coverslips containing the cells were then immersed in fixative solution (3% PFA and 2% glutaraldehyde) and stored at 4˚C until further processing for imaging.

For Transmission Electron Microscopy (TEM): Samples were fixed in Karnovsky's fixative made with 2% paraformaldehyde and 2.5% glutaraldehyde in 0.1 M phosphate buffer (PB). All subsequent steps were carried out in a Pelco Biowave Pro laboratory microwave (MW) (Ted Pella Inc., Redding, CA) using the 2 min on, 2 min off, and 2 min on cycle. The samples were fixed in the MW at 170 Watts and later rinsed in 0.1 M PB. Following post-fixation in 1% osmium tetroxide reduced with 8% potassium ferrocyanide in 0.1 M PB, the cells were rinsed with PB and en bloc stained with 1% uranyl acetate. After rinsing the samples with water, they were infiltrated with water-soluble Durcupan resin according to the manufacturer's instructions. Samples were hardened for 48 h and ultrathin sections were collected using a Leica UC6 ultramicrotome (Leica Microsystems, Wetzlar, Germany). The sections were imaged at 80 kV

in Hitachi 7800 microscope (Hitachi High-Tech in America, Schaumburg, IL) using an AMT XR-81 camera (Advanced Microscopy Techniques, Inc., Woburn, MA).

For Scanning Electron Microscopy (SEM): Cells on silicon chips were fixed in Karnovsky's fixative and post-fixed with reduced osmium tetroxide as mentioned in the TEM method. After washing the samples in water, the cells were dehydrated in ethanol and dried using a Baltech critical point dryer. The chips were mounted onto SEM stubs using silver paint. The cells were sputter coated with 8nm of iridium (Ir) using the Quorum Q300T sputter coater (Quorum Tech, UK). SEM images were captured using Hitachi SU 8000 microscope (Hitachi High-Tech in America, Schaumburg, IL) Subsequent post-fixation with 1% OsO4 was performed with microwave irradiation (Pelco 3451 microwave processor, in cycles of 2 min on, 2 min off, 2 min on at 250 W under 15 in. (Hg vacuum; Ted Pella). Specimens were dehydrated in a graded ethanol series for 1 min under vacuum. Samples were then dried to a critical point in a Bal-Tec cpd 030 drier (Balzer, Bal-Tec AG, Balzers, Liechtenstein). Cells were then coated with 75 A˚ of iridium in an IBS ionbeam sputter (South Bay Technology, Inc., San Clemente, CA.) Samples were imaged on a Hitachi SU-8000 SEM (Hitachi, Pleasantown, CA).

## Identification of parasitophorous vacuole (PV)

Trophozoite enriched iRBC and resting NK cells were washed with PBS. The iRBC were stained with 1.5 μM of PKH26 membrane dye for 3 min at RT. Cells were then washed once with RPMI with 10% FBS and once with PBS. The iRBC were then stained with cell proliferation dye eFluor 670 (2.5 μM) for 5 min at 37˚C. Cells were then washed twice with media containing serum. Resting NK cells were stained with 2 μM of PKH67 membrane dye for 5 min at RT. Cells were then washed twice with media containing serum. Stained iRBC ($2\times10^5$) either alone or mixed with stained resting NK cells ($1\times10^6$) were added to a 96-well plate, in the presence of anti-RBC rabbit antibody (2 μg/mL) and incubated for 1 h. Cells were then washed and analyzed by flow cytometry (X-20, BD Biosciences), followed by data analysis with FlowJo (FlowJo, LLC). PKH67-PKH26-eFluor+ cells are considered as parasitophorous vacuoles.

## Generation of EXP2-GFP 3D7

The EXP2 locus was C-terminally tagged with monomeric GFP using the two-vector Cas9 genome editing system as described [60]. For sgRNA creation, a guide sequence (5′ – `gttgaagaagaagatgccag`-3′) derived from the last exon of EXP2 was cloned into the BtgZ1 site of the pL6 vector. For homology-arms generation, EXP2 nucleotides 959–1337 were PCR amplified to create the 5'-homology arm and 515 nucleotides immediately following the stop codon were PCR amplified to create the 3'-homology arm. DNA encoding EXP2 nucleotides 1,338 to 1,400 was mutated to retain amino acid encoding but destroy sgRNA binding (a shield mutation). A thrombin cut site, monomeric GFP, 3xFLAG tag, octahistidine sequence, and stop codon were cloned between the 5' and 3'-homology arms. The 5'-homology arm, codon-optimized EXP2-thrombin-GFP-3xFLAG-octahistidine, stop codon and 3'-homology arm were cloned into the pL6 Not1 and AflII restriction sites to create pL6-EXP2-GFP.

Uninfected RBC were washed and electroporated with pUF1-Cas9 and pL6-EXP2-GFP. Transfected uninfected RBC were mixed with trophozoite stage enriched 3D7 iRBC and cultured for two days without selection in RPMI 1640 with 2 mM L-glutamine, 25 mM HEPES, 0.5% Albumax, 0.26% sodium bicarbonate, 10 μg/mL gentamicin, and 50 μg/mL hypoxanthine. After 2 days, 2 mL fresh 50% uninfected RBC and media with 1.5 μM DSM1 (MR4) and 2 nM WR99210 (both from Jacobus Pharmaceuticals) was used for selection. After 3–4 weeks, the 3D7-EXP2-GFP iRBC were isolated by limited dilution cloning. The presence of EXP2-GFP was confirmed by western blotting, PCR and imaging.

## Phalloidin and PI staining of iRBC

Trophozoite enriched 3D7 iRBC were washed with PBS and stained with 2.5 μM eFluor 670 for 5 min in PBS at 37˚C. Cells were then washed twice in RPMI 1640 with 10% FBS. Enriched eFluor 670 stained 3D7 iRBC ($2\times10^5$) either alone or mixed with resting NK cells ($1\times10^6$) were added to a 96-well plate, in the presence of antibodies (anti-RBC rabbit antibody at 2 μg/mL or 4 μg/mL) and incubated at 37˚C for 1–3 h. Cells were washed and stained with Pe-Cy7 CD235a and FITC-LFA-1 for 15 min on ice. Cells were washed and stained with 165 nM AF405-phalloidin for 10 min at RT, followed by 1.25 μg/mL PI for 5 min. Flow cytometry was performed on LSRFortessa X20 cell analyzer (BD Biosciences), and data analyzed with FlowJo (FlowJo, LLC). For immunofluorescence analysis, the cells were transferred to an 8 well chamber and images were collected on a LSM 780 confocal laser microscope (Carl Zeiss, Germany). Similarly, enriched 3D7-EXP2-GFP iRBC were stained with 1.5 μM of PKH26 membrane dye, followed by 2.5 μM of eFluor 670. Enriched PKH26 and eFluor 670 stained EXP2-GFP 3D7 iRBC either alone or mixed with resting NK cells were added to a 96-well plate, in the presence of antibodies (anti-RBC rabbit antibody at 2 μg/mL) and incubated at 37˚C for 3 h. Images were then collected on LSM 780 confocal laser microscope. Images were acquired using the Zen software (Carl Zeiss).

## Isolation of PV

iRBC were stained with 1.5 μM of PKH26 (or PKH67 or CellVue Claret) membrane dye, followed by 2.5 μM eFluor 670 (or 5 μM eFluor 450) respectively. Stained iRBC were permeabilized first by non-ionic detergent chemical disruption (0.0125% saponin for 3 min), washed once in PBS, and this was followed by mechanical disruption (31G needle). For PV released after NK cell mediated lysis of iRBC, resting NK cells were stained with 5 μM of eFluor 450 followed by two washes with media containing serum. Stained iRBC mixed with eFluor 450 stained resting NK cells were added to a 6-well plate, in the presence of 4 μg/mL rabbit anti-RBC antibody and incubated for 1 h. Cells (after manual disruption as well as after NK cell mediated lysis) were resuspended in 0.5% Albumax and filtered through a 35 μm filter (Corning) immediately before cell sorting. Vacuoles were sorted using BD FACSAria TM Fusion, and gating was performed using the BD FACSDiva TM software (BD Biosciences). Samples were sorted with a sheath pressure of 70 psi, a 70 μm nozzle, and a maximum event rate of 10,000 events per second. Membrane dye negative, eFluor+ cells (PV) were collected, washed and used for experiments.

## Antibody staining

Enriched iRBC, uninfected RBC and isolated PV after saponin lysis were incubated in the presence of different concentrations of antibodies (US IgG, Mali IgG or anti-RBC IgG). Antibody binding to the cells was detected with BV421 secondary antibodies. Samples were washed, resuspended in PBS and analyzed on a FACS LSRII flow cytometer (BD Biosciences). Data analysis was performed with FlowJo software (FlowJo, LLC).

## Phagocytosis assay

iRBC were stained with 1.5 μM of PKH26 (or CellVue Claret) membrane dye, followed by 2.5 μM eFluor 670 (or 5 μM eFluor 450). PV from membrane dye-negative eFluor+ cells were isolated after flow sorting. Isolated PV ($1.5\times10^5$) either in presence or absence of antibodies (rabbit anti-RBC antibody at 100 μg/mL, US IgG at 80 μg/mL or Mali IgG at 80 μg/mL) were added to 24 well plates already seeded with monocytes ($2\times10^5$) and incubated at 37˚C for 1 h.

For certain donors, after staining iRBC with CellVue Claret membrane dye and eFluor 450, the cells were further stained with 4 µg/mL pHrodo (Thermo Fisher, P36011). The pHrodo stained iRBC underwent cell sorting and PV from membrane dye negative eFluor+ cells were collected. These PV were then incubated with monocytes in presence or absence of antibodies. Monocytes were washed, stained with LIVE-DEAD stain and then flow cytometry was performed on them on LSRFortessa X20 cell analyzer (BD Biosciences), and data analyzed with FlowJo (FlowJo, LLC).

### Time-lapse imaging

For NK cell-mediated lysis, NK cells and iRBC were washed twice with PBS before labeling with different dyes. iRBC were stained with 2.5 µM eFluor 670 for 5 min in PBS at 37˚C. NK cells were stained with 5 µM eFluor 450 for 8 min in PBS at 37˚C. Cells were then washed twice and resuspended in RPMI 1640 containing 0.5% Albumax-II in the absence of Phenol Red. Cells were added in 8-well Lab-Tek I Chambered cover glass (Nunc) and allowed to settle for 10 min. 1.25 µg/mL PI was added to the wells. Similarly, iRBC were stained with 1.5 µM of PKH26 membrane dye, washed and then stained by 2.5 µM eFluor 670. For imaging, cells were resuspended in RPMI 1640 containing 0.5% Albumax-II in the absence of Phenol Red. Cells were added in 8-well Lab-Tek I Chambered cover glass (Nunc) and allowed to settle for 10 min. 165 nM AF405-phalloidin was added to the wells.

For monocyte experiments, iRBC were stained with 1.5 µM of PKH67 membrane dye, washed and then stained by 2.5 µM eFluor 670. PV were isolated as described above. On the day of the experiment, monocytes were thawed, washed and stained with 5 µM eFluor 450. Stained monocytes were dropped onto 8-well Lab-Tek I Chambered cover glass and allowed to settle for 2 h at 37˚C. The chambers were washed once with PBS and then PV were added and allowed to settle for 10 min. 1.25 µg/mL PI was added to the wells. Imaging was performed with a Zeiss LSM 780 confocal microscope while maintaining incubation condition at 37˚C, 5% CO2, in a humidified chamber. Images were acquired at 60 s interval for 5 h. Time-lapse image stacks were imported into the Imaris software.

### Statistical analysis

The statistical tests used are specified in the figure legends. Statistical tests were computed using GraphPad Prism version 9.0 (http://www.graphpad.com/scientific-software/prism/)

### Supporting information

**S1 Fig. Flow cytometry profiles of EJ cells and RBC.** (**A**) Flow cytometry profiles of cell surface staining using Rabbit anti-RBC polyclonal antibody in uninfected and 3D7-infected RBC. (**B**) Flow cytometry profiles of cell surface staining of Cas9 control, ICAM4 KO and CD58 KO EJ cells for ICAM4 and CD58, showing the loss of ICAM4 and CD58 in the respective knockout cells.
(PDF)

**S2 Fig. Gating to identify iRBC and a control for damage to iRBC in the absence of NK cells.** (**A**) Gating of eFluor 670+ glycophorin A (CD235a)+ iRBC and LFA-1+ NK cells to determine damage to iRBC with Phalloidin-AF405 and to the parasite with PI. (**B**) Same analysis as in (A) of iRBC incubated for 3 hours with anti-RBC IgG in the absence of NK cells. (**C**) Representative experiment performed as in Fig 2C performed in the absence of NK cells.
(PDF)

**S3 Fig. Serial killing of *P.f.* in iRBC by a single NK cell.** The experimental setup is identical to that shown in Fig 3. eFluor 450-stained primary NK cells (blue) were incubated with eFluor 670-stained iRBC (red) at a ratio of 1:3 in the presence 2 μg/ml anti-RBC IgG. PI was added at the start of image recording. The 12 time frames span 52 minutes. Numbers at the upper left corner of each panel indicate the time in minutes relative to the first NK–iRBC contact, which was set as minute 0. A ghost iRBC (labeled *G* in the first panel, -7 minutes) was contacted by an NK cell (marked with an asterisk) at time 0. This ghost iRBC included a PI-negative PV and was adjacent to an intact iRBC target (marked *T*). Activation of the NK cell in contact with the iRBC ghost was detected after 2 minutes (t = 2) through enhanced motility and plasma membrane extensions. PV damage was observed 3 minutes later (yellow arrow). One minute later (t = 4 min), the NK cell covered the iRBC target and began to move away. The ghost iRBC membrane was dragged away by the NK cell, leaving a free PI+ PV behind (t = 9). The motile NK cell contacted several other iRBC (beginning at t = 9). PI+ PV appeared at t = 20, 27, and 45 (yellow arrows). Note the presence of a PV in a ghost iRBC (bottom right) that remained PI-negative for at least 52 minutes.
(PDF)

**S4 Fig. Electron microscopy images of NK cells, uninfected RBC, iRBC, ghost iRBC, and PV.** (**A**) TEM image of an NK cell and three uninfected RBC, which had been co-incubated in the presence of 2 μg/ml anti-RBC antibody for 3 hours. Scale bar is 5 μm. (**B**) TEM image of three NK cells in contact with two iRBC during incubation with 2 μg/ml anti-RBC antibody. Two NK cells (lower left and lower right) that flank the same iRBC were polarized, as seen by nucleus positioned at the back of the cell, away from the immunological synapse with the iRBC target. The lower left NK cell formed synapses with 2 iRBC. The NK cell at the top, apparently less polarized, may have been detaching from a ghost iRBC. Scale bar is 5 μm. (**C**) SEM image of an NK cell and an iRBC co-incubated in the absence of anti-RBC antibody. Scale bar is 5 μm. (**D**) SEM images of NK cells in contact with iRBC after incubation with 2 μg/ml anti-RBC antibody. The iRBC in the first image (top panel) appears to have retained some rigidity. In the lower panel, loss of iRBC rigidity suggests that damage by the NK cell had occurred. Scale bars are 5 μm. (**E**) SEM image of a PV at the center, which was surrounded by four NK cells and three iRBC ghosts (arrowheads), taken from a sample obtained after coincubation of NK cells and iRBC in the presence of 2 μg/ml anti-RBC antibody. Scale bar is 5 μm.
(PDF)

**S5 Fig. Gating used to separate iRBC from NK cells and identify free PV.** EXP2-GFP + iRBC were stained with eFluor 670 and membrane dye PKH26. NK cells were stained with PKH67. iRBC were gated away from NK cells first by forward and side scatter, and then gated for eFluor670+ PKH67-negative cells, followed by gating the PKH26-negative eFluor670+ population consisting of free PV.
(PDF)

**S6 Fig. Preparation of PV extracted from iRBC and staining with anti-RBC IgG.** (**A**) RBC infected with 3D7 PfEXP2-GFP were stained with membrane dye PKH26 and eFluor 670 and treated with 0.0125% saponin for 3 minutes, followed by a wash and mechanical disruption of the iRBC plasma membrane through a 31-gauge needle. (**B**) Samples were sorted by flow cytometry for PKH26-negative eFluor 670+ particles. (**C**) Mean fluorescence intensity (MFI) of uninfected RBC (uninfected RBC), iRBC, and PV stained with different concentrations of anti-RBC IgG. The first set of points represent samples stained with secondary Ab only (sec only).
(PDF)

**S7 Fig. Gating strategy for eFluor 450+ and pHrodo+ monocytes.** Monocytes previously incubated with PV were stained with live-dead stain and gated for live monocytes and further gated for either eFluor 450+ or pHrodo+ monocytes.
(PDF)

**S1 Movie. Video of Fig 3A.**
(MP4)

**S2 Movie. Video of Fig 3B.**
(MP4)

**S3 Movie. Video of S3 Fig.**
(MOV)

**S4 Movie. Video of Fig 5C.**
(MP4)

**S5 Movie. Video of Fig 5D.**
(MOV)

**S1 Data. Primary data that supports the numerical graphs shown in the manuscript.**
(XLSX)

## Acknowledgments

Research at the National Institute of Allergy and Infectious Diseases (NIAID), National Institutes of Health (NIH), was supported by the Division of Intramural Research, NIAID. We thank L. Krymskaya for cell and vacuole sorting (Cell Sorting and Analysis Core, Laboratory of Immunogenetics), J. Brzostowski for help with imaging (Twinbrook Imaging Facility, Laboratory of Immunogenetics), D. Dorward and E. Fischer for electron microscopy (Microscopy Unit, Rocky Mountain Laboratories, NIAID). DSM1 (MRA-1161) was obtained through MR4 as part of the BEI Resources Repository, NIAID, NIH. WR99210 was a generous gift from Jacobus Pharmaceuticals.

## Author Contributions

**Conceptualization:** Gunjan Arora, Eric O. Long.

**Funding acquisition:** Sanjay A. Desai, Eric O. Long.

**Investigation:** Padmapriya Sekar, Sumati Rajagopalan, Estela Shabani, Usheer Kanjee, Marc A. Schureck, Gunjan Arora, Mary E. Peterson.

**Methodology:** Sumati Rajagopalan, Manoj T. Duraisingh, Sanjay A. Desai.

**Project administration:** Eric O. Long.

**Resources:** Boubacar Traore, Peter D. Crompton, Manoj T. Duraisingh.

**Supervision:** Sumati Rajagopalan, Sanjay A. Desai, Eric O. Long.

**Writing – original draft:** Padmapriya Sekar, Sumati Rajagopalan, Eric O. Long.

**Writing – review & editing:** Padmapriya Sekar, Sumati Rajagopalan, Eric O. Long.

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
