## [Decision Letter · Decision Letter 0]

7 Sep 2023

Dear Dr. Long,

Thank you very much for submitting your manuscript "NK cell-induced damage to P. falciparum-infected erythrocytes requires ligand-specific recognition and releases parasitophorous vacuoles that are phagocytosed by monocytes in the presence of immune IgG" for consideration at PLOS Pathogens. As with all papers reviewed by the journal, your manuscript was reviewed by members of the editorial board and by several independent reviewers. In light of the reviews (below this email), we would like to invite the resubmission of a revised version that takes into account the reviewers' comments.

The reviewers of your manuscript noted that your results are informative and important with some novel elements. On the basis of their reviews we believe your paper is potentially acceptable for publication in PLOS Pathogens. However, the reviewers did raise several important issues that you need to address and have a number of suggestions to improve your manuscript. Therefore we are returning the manuscript to you for revision. Please address these points as appropriate and provide a response to each comment.

We cannot make any decision about publication until we have seen the revised manuscript and your response to the reviewers' comments. Your revised manuscript is also likely to be sent to reviewers for further evaluation.

Sincerely,

James G. Beeson, MBBS, PhD

Academic Editor

PLOS Pathogens

Dominique Soldati-Favre

Section Editor

PLOS Pathogens

Kasturi Haldar

Editor-in-Chief

PLOS Pathogens

orcid.org/0000-0001-5065-158X

Michael Malim

Editor-in-Chief

PLOS Pathogens

orcid.org/0000-0002-7699-2064

The reviewers of your manuscript noted that your results are informative and important with some novel elements. On the basis of their reviews I believe your paper is potentially acceptable for publication in PLOS Pathogens. However, the reviewers did raise several important issues that you need to address and have a number of suggestions to improve your manuscript. Therefore we are returning the manuscript to you for revision. Please address these points as appropriate and provide a response to each comment.

Reviewer's Responses to Questions

**Part I - Summary**

Reviewer #1: In this beautifully written paper by Sekar et al, the authors demonstrate how NK cells are able to both bind opsonized Pf-infected RBCs and what happens to parasitic contents following NK-cell mediated ADCC. In a series of elegant experiments combining both antibody blocking experiments and gene ablation in erythrocyte precursors, the authors show that red cell expression of ligands CD58 and ICAM-4 are critical for adhesion to NK cells via their receptors CD2 and aMB2, respectively. Interestingly, adhesion required CD16-dependent signals, but NK cells were also able to discriminate between infected and uninfected red blood cells even in the presence of polyclonal rabbit anti-RBC IgG. The authors then show that NK-mediated ADCC first resulted in damage to the IRBC plasma membrane followed by release of intact PV, and that these PV could be subsequently opsonized by immune IgG and phagocytosed by monocytes. Overall I found the manuscript to be fascinating, with many thoughtful experiments describing these important mechanisms of action of NK cells. I do have a few comments below that if addressed will assist in interpretation of the study findings.

Reviewer #2: Sekar et al. have in this manuscript aimed to study how NK cells provide protection during malaria. They do this by identifying key receptor interactions between NK cells and iRBC and mechanisms for NK-mediated iRBC damage/killing. They have further assessed the role of antibodies in this process and how those antibodies work together with monocytes to clear opsonized parasite material. To obtain this data they have used elegant in vitro systems and tools to visualize and measure the different processes. Overall, the manuscript is well-written and comprehensive and presents novel relevant data.

**Part II – Major Issues: Key Experiments Required for Acceptance**

Reviewer #1: - My only major comment is regarding the specificity for the interaction/conjugate formation between infected and uninfected RBCs. In Figure 1a-C, the authors show that, in the presence of polyclonal anti-RBC IgG, NK cells only form conjugates with iRBC and not uninfected RBCs. This result is surprising to me. The authors speculate in the discussion that iRBC may be more rigid, favoring NK cell binding; alternatively, that the distribution of ICAM-4 might be different on iRBC vs. uninfected RBC and that this could favor aMB2 binding. Did the authors compare ICAM-4 expression between uninfected and infected RBCs? Is that something that could be shown? Alternatively, are the ligands bound by polyclonal antiRBC IgG serum different between infected and uninfected RBCs?

- Relatedly - In the discussion, 2nd paragraph, the authors state, “This selectivity occurs even when uninfected RBC ad iRBC are coated with the same polyclonal antiRBC IgG serum [6].” I could not find evidence for this in the Arora et al ELIFE paper. Should that reference be removed since this result seems specific to the present paper?

- In the forward genetic screening experiments, the authors noted that NK cell lysis of uninfected EJ cells opsonized with anti-RBC IgG serum was comparable to NK cell lysis of infected RBCs. I am a little puzzled by this given the above observation of the specificity of killing in the prior experiments to iRBCs. They did note some differences in this model (e.g., blocking of aMB2 did not abrogate lysis, although gene ablation of ICAM-4 did). Did the authors try infecting the EJ cells with parasites to see how this might impact expression of cell surface markers and/or NK cell lysis? If not, possibly include this in the discussion as a potential limitation?

Reviewer #2: No additional experiments are required but there are several questions regarding the experiments that need clarification. These are added in the next section (Part III-Minor issues).

**Part III – Minor Issues: Editorial and Data Presentation Modifications**

Reviewer #1: - In the results, the authors show data that the parasitophorous vacule is opsonized by antibodies. Could the authors speculate in the discussion what the targets for these antibodies are? Do the authors have any data that these antibodies might accumulate with increasing exposure to Pf? (e.g. any correlations with age?)

- The authors show that these opsonized PV are then phagocytosed by monocytes. Did the authors determine whether these opsonized PV could be destroyed by other mechanisms? (e.g., NK-mediated ADCC? ADCP by neutrophils?) I wasn’t entirely sure why monocyte phagocytosis was speculated to be the preferred route of elimination of PV; clarification here or in the discussion would help the reader.

- it’s not very clear how many cells were used in each assay; could be helpful detail

Figures

• 1A:

• The authors could be more explicit with the opsonizing conditions in the figure legend; the only visual difference between opsonizing and non-opsonizing conditions is filled-in shapes, many of which overlap. Filled-in dots/squares could be labeled more informatively, like “iRBC + Ab” instead of just “iRBC.”

• A brief description of the setup, markers, and determination of cell conjugates used in the two-color flow assay may add helpful contextual information to readers.

• 1B:

• What do the three colors represent - three different donors? What time post co-incubation were cell-cell conjugates quantified? (pls add to legend)

• 1E + 1F:

• Although it’s clear from the text that the experimental setup for 1F was the same as 1E aside from using differentiated EJ cells, I think that could be clearer in the figures since they look nearly identical side-by-side

• 1G + 1H:

• Could the authors include wild-type / non Cas9-transfected EJ cells in these figures?

Figure 8

• 8B:

• The text specifies that primary monocytes and NK cells were used from multiple different donors: was each experiment done using cells from autologous donors? Would clarify that these donors were malaria-naïve since plasma is from malaria-exposed

Reviewer #2: Minor questions, comments, and clarifications asked for.

1) Is there a reason the NK cell conjugate assay (Figure 1A-C) does not work with the receptor blocks or why those experiments were not added?

2) It is interesting that so much of the NK-mediated killing remains despite blocking the ICAM4 and CD58 receptor-ligand interactions. Do the authors think there could be additional receptor-ligand pairs or is the remaining interaction by CD16-Ab alone?

3) Although using the rabbit-anti-RBC standardizes coating of RBC, independent of parasites, it also leaves me wondering a bit how anti-RBC + iRBC + NK cells would correlate with for example Mali plasma + iRBC + NK cells. In figure 7 it seems there is very little binding of Mali IgG to iRBC, so would the NK cells in vivo be nearly as active/functional as what is shown here? Could the much larger RBC coating when using anti-RBC Abs influence for example the killing/breaking mechanism of the RBCs.

4) Another Q following on the previous question is that by using rabbit anti-RBC, is there any influence on Fc-mediated functions compared to if human antibodies would have been used?

5) The monocyte uptake assay is nice and the data is convincing. However, in blood the majority of cells would be neutrophils, which also express high levels of CD16 and are known to both phagocytose and use other mechanisms for killing pathogens and are proposed to play an important role (e.g. PMID: 34413459). Any thoughts on their role in this system/process? Perhaps add a line of discussion/limitation.

6) The results are mainly discussed in the context of adaptive NK cells and touch upon CD16low NK cells. But in the assays, total NK cells (CD56+) were used.

Minor comments

-Page 9 line 8: cells is written twice “cells cells”

-Page 12: There are two “is is” following each other.

-In figure 2B, it says Phalloidin 450, should be 405.

-When discussing hemozoin and its stimulation, it could be worth referencing this paper: PMID: 17261807.

-When introducing/discussing tolerance antibody-monocyte interplay, it could be worth to include this paper: PMID: 35443186

PLOS authors have the option to publish the peer review history of their article (what does this mean?). If published, this will include your full peer review and any attached files.

Reviewer #1: No

Reviewer #2: No
---

## [Editor Report · Decision Letter 1]

24 Oct 2023

Dear Dr. Long,

We are pleased to inform you that your manuscript 'NK cell-induced damage to P.falciparum-infected erythrocytes requires ligand-specific recognition and releases parasitophorous vacuoles that are phagocytosed by monocytes in the presence of immune IgG' has been provisionally accepted for publication in PLOS Pathogens.

Best regards,

James G. Beeson, MBBS, PhD

Academic Editor

PLOS Pathogens

Dominique Soldati-Favre

Section Editor

PLOS Pathogens

Kasturi Haldar

Editor-in-Chief

PLOS Pathogens

orcid.org/0000-0001-5065-158X

Michael Malim

Editor-in-Chief

PLOS Pathogens

orcid.org/0000-0002-7699-2064
---

## [Editor Report · Acceptance letter]

30 Oct 2023

Dear Dr. Long,

We are delighted to inform you that your manuscript, "NK cell-induced damage to *P.falciparum*-infected erythrocytes requires ligand-specific recognition and releases parasitophorous vacuoles that are phagocytosed by monocytes in the presence of immune IgG," has been formally accepted for publication in PLOS Pathogens.

Best regards,

Kasturi Haldar

Editor-in-Chief

PLOS Pathogens

orcid.org/0000-0001-5065-158X

Michael Malim

Editor-in-Chief

PLOS Pathogens

orcid.org/0000-0002-7699-2064